# Pinealectomy-Induced Neuroinflammation Varies with Age in Rats

**DOI:** 10.3390/ijms26168093

**Published:** 2025-08-21

**Authors:** Dimitrinka Atanasova, Desislava Krushovlieva, Pavel Rashev, Milena Mourdjeva, Despina Pupaki, Jana Tchekalarova

**Affiliations:** 1Institute of Neurobiology, Bulgarian Academy of Sciences, 1113 Sofia, Bulgaria; didiatanasova7@gmail.com (D.A.);; 2Department of Anatomy, Faculty of Medicine, Trakia University, 6003 Stara Zagora, Bulgaria; 3Institute of Biology and Immunology of Reproduction “Acad. Kiril Bratanov”, Bulgarian Academy of Sciences, 1113 Sofia, Bulgariamilena_mourdjeva@abv.bg (M.M.); poupaki_desi@abv.bg (D.P.)

**Keywords:** aging, pinealectomy, inflammation, pAkt1, NF-kB, glia

## Abstract

It is widely accepted that chronic inflammation constitutes a significant mechanism that promotes the biological aging process. The pineal gland is regarded as being closely related to the control of the “life clock”. The present study aimed to determine the inflammation associated with pinealectomy in the rat hippocampus and to investigate the extent to which age stage impacts the severity of this inflammation. We evaluated the expression of the Akt/NF-kB signaling pathway in neurons and gliosis level in the dorsal hippocampus (dHipp) of rats subjected to sham surgery or pinealectomy at 3, 14, or 18 months of age. The assessment was conducted using immunohistochemistry. Removal of the pineal gland resulted in significant, region-specific increases in NF-kB expression in neurons of the dHipp in the youngest and middle-aged groups. However, the change in expression of the phosphorylated form of Akt (pAkt1) in neurons went in the opposite direction in these two age groups, and there were also regional differences. Pinealectomy triggered microgliosis in both young and old rats, but middle-aged rats were resistant to microglia activation. Conversely, astrogliosis was observed in young adult and middle-aged groups with melatonin deficiency in certain regions of the dHipp. It is noteworthy that young adult rats demonstrated the highest degree of vulnerability to inflammation associated with the loss of melatonin as a hormone. In contrast, middle-aged rats with pinealectomy exhibited a complex and partial adaptive response. These findings emphasize the dynamic and age-dependent nature of neuroinflammation following pinealectomy, underscoring the developmental stage as a critical determinant of inflammatory susceptibility.

## 1. Introduction

Aging, a universal and intrinsic phenomenon, is a shared experience that impacts every animal, including humans [1]. While age-related diseases and degenerative conditions can vary among individuals, they exhibit strikingly similar patterns [2]. Chronic inflammation is a significant factor that accelerates the aging process and contributes to various health issues. Age-related triggers of multiple inflammatory signaling pathways in the brain could lead to altered neuronal function, reduced plasticity, and increased susceptibility to neurodegenerative diseases. As individuals age, microglia become primed, resulting in heightened reactivity to stimuli, and chronic inflammation significantly accelerates aging and contributes to various health issues. Several key players in inflammatory responses are affected by aging, including the brain’s resident immune cells—specifically microglia and astrocytes—the transcription factor NF-kB (nuclear factor kappa-light-chain-enhancer of activated B cells), and the Akt pathway (also known as the Protein Kinase B pathway) [3,4,5,6]. Studies support the assumption that different factors influencing premature aging or delaying this process converge through the NF-kB signaling pathway [7]. In general, the consequences of long-term neuroinflammation could result in a positive feedback loop of closely related devastating events as follows: (1) cytokine-induced impaired synaptic functions due to changed neurotransmitter release and synaptic plasticity failure; (2) neuronal loss due to increased oxidative stress and concomitant apoptosis; (3) cognitive decline and progressive neurodegeneration associated with diseases such as Alzheimer’s, Parkinson’s, and multiple sclerosis [8,9,10].

Aging and melatonin are profoundly intertwined, as melatonin is essential for age-associated processes that worsen over time, such as the control of circadian rhythms, sleep quality, antioxidant defense enhancement, and neuroprotection [11]. Understanding this connection underscores the significance of melatonin in maintaining our health and vitality as we age. Melatonin levels decrease with age, starting as early as middle age, and this decline can have significant implications for overall health and well-being [12,13]. The hormone impacts the suprachiasmatic hypothalamic nucleus via a feedback mechanism that synchronizes internal time with the external light–dark cycle [14]. In aging, this crucial relationship is weakened, and the brain’s “clock” becomes less responsive to melatonin and vice versa [15]. The deficit in melatonin resulting from aging leads to a high risk of insomnia due to a predisposition to phase shifts and sleep fragmentation. Aging may reduce the expression or sensitivity of MT1 and MT2 melatonin receptors, affecting signal transduction [16].

Melatonin has been demonstrated to exert a multifaceted effect on the immune system, functioning both as a target and a direct agent [17]. It has been demonstrated that the pineal gland can be targeted by inflammatory signaling molecules, such as interferon-gamma [18] and tumor necrosis factor-alpha (TNF-α) [19]. This process has been shown to affect melatonin synthesis as a hormone. The protective activity of the hormone was shown to be exerted via the pro-survival PI3K/Akt and SIRT1 pathways or inflammation-related TLR4/NF-κB and MAPK [20]. Moreover, melatonin has been demonstrated to interact with microglia and mitigate their activation in the brain [21]. Studies support the presumption that melatonin deficiency can lead to increased neuroinflammation [22,23,24].

Our recent studies and various reports indicate that the relationship between melatonin deficiency and aging is complex [25,26,27,28,29,30]. Generally, pinealectomy in younger animals has been shown to accelerate aging, which results from increased oxidative stress, dysfunction in metabolic processes, cognitive decline, and impaired emotional well-being. Conversely, melatonin deficiency in older animals does not appear to influence the pace of aging. However, findings regarding middle-aged rodents with melatonin deficiency have been inconsistent, with some studies suggesting protective effects while others point to dysfunction. This inconsistency underscores the importance of this age group in understanding melatonin’s role and suggests that further research is warranted. In the present study, we evaluated inflammatory markers of aging—specifically, active microglial cells and astrocytes, NF-kB, and Akt signaling—using immunohistochemistry in different regions of the rat dorsal hippocampus (dHipp) across three age groups: young adults, middle-aged, and old rats. We hypothesized that melatonin deficiency resulting from pinealectomy would exacerbate these neuroinflammatory markers in an age-dependent manner. Our results confirmed that young adulthood is the most vulnerable stage during which melatonin dysfunction can accelerate aging through increased neuroinflammation. In contrast, the neuroinflammatory response associated with pinealectomy in middle-aged rats was more complex, suggesting the involvement of additional mechanisms that interact with this process.

## 2. Results

### 2.1. Effects of Pinealectomy and Aging on Phosphorylated Form of Akt (pAkt1) Expression in Different Hippocampal Regions

Pinealectomy significantly altered pAkt1 protein levels in several hippocampal regions, with interaction between Age and Surgery contributing to these changes.

Immunohistochemical detection of pAkt1 protein revealed distinct age-dependent and surgery-related patterns of expression across multiple hippocampal subregions, including the molecular (MoDG), granular (GrDG), and polymorphic (PoDG) layers of the dentate gyrus (DG), as well as the CA3c, CA3a, and CA1 areas of the Ammon’s horn (Figure 1).

In 3-month-old sham-operated rats (Figure 1(A1–A5)), pAkt1 immunoreactivity was pronounced across all regions, with dense cytoplasmic and nuclear labeling observed in neurons of the CA1 and CA3a subfields, as well as in cells spanning all layers of the dentate gyrus. This pattern indicates a physiologically high level of pAkt1 expression in the young adult dHipp. In contrast, age-matched pinealectomized rats (Figure 1(B1–B5)) showed a visible reduction in pAkt-positive cells, particularly in CA1, CA3a, and MoDG, consistent with the quantitative data indicating a significant pinealectomy-induced suppression of pAkt1 expression at this age. At 14 months, sham animals (Figure 1(C1–C5)) displayed moderate pAkt1 immunoreactivity, most apparent in the CA3a, CA1, and PoDG regions. Notably, pinealectomized rats of the same age (Figure 1(D1–D5)) exhibited markedly stronger labeling, especially in CA3c and the dentate gyrus, where all three layers (MoDG, GrDG, PoDG) contained densely labeled cells. This finding aligns with the observed increase in pAkt1-positive cell density following pinealectomy at this intermediate age. In 18-month-old sham-operated animals (Figure 1(E1–E5)), pAkt1 expression remained moderate and was relatively well-distributed across hippocampal subfields, though staining intensity appeared lower in CA1 and MoDG compared to younger rats. Strikingly, 18-month-old pinealectomized animals (Figure 1(F1–F5)) demonstrated robust and widespread pAkt1 immunoreactivity. Dense labeling was observed throughout CA3c and all dentate layers, indicating a substantial age-related upregulation of pAkt1 signaling following chronic melatonin deficiency.

To investigate the basal cellular compartmentalization of Akt signaling, we performed double immunofluorescence staining for pAkt1 and canonical neuronal and glial markers on brain sections from the cornu ammonis area 1 (CA1) of 3-month-old rats that underwent sham surgery. Our findings revealed that pAkt1 immunoreactivity was strong and localized specifically to the pyramidal cell layer (Figure 2). NeuN-positive pyramidal neurons exhibited intense cytoplasmic and perisomatic labeling that occasionally extended into proximal dendrites. This demonstrated clear colocalization with NeuN in the merged images.

In contrast, we did not detect pAkt1 in the glial compartments (Figure 2). GFAP-positive astrocytes, which traverse the neuropil of the strata oriens and radiatum, lacked pAkt1 immunoreactivity. Merged images showed no spatial overlap between GFAP and pAkt1. Similarly, Iba1-positive microglia displayed no pAkt1 signal or co-localization.

Quantitative analysis of pAkt1-immunoreactive cells confirmed a dynamic, region-specific modulation of pAkt1 expression with aging and pinealectomy in the dHipp (Figure 3). In the MoDG, main effects of Surgery [F(1,72) = 52.135, *p* < 0.001] was observed. Pinealectomy induced a significant increase in pAkt1 expression in 14-month-old (*p* < 0.001) compared to sham controls, whereas it caused a significant decrease in 3-month-old rats compared to matched controls (*p* = 0.022) (Figure 3A).

In the GrDG, no significant main effect of Surgery was detected (*p* = 0.816) but a main effect of Age [F(2,197) = 7.664, *p* < 0.001] and a significant Surgery × Age interaction [F(2,197) = 19.943, *p* < 0.001] was observed. Post hoc analysis indicated a significant age-dependent decrease in pAkt levels among sham-operated rats (*p* < 0.001, 14- and 18-month-old rats vs. 3-month-old rats) (Figure 3B). Furthermore, pinealectomy significantly increased pAkt1 expression in 14-month-old rats (*p* = 0.010) compared to their matched sham controls. In contrast, melatonin deficit reduced pAkt1 expression in 3-month-old rats compared to 3-month-old sham rat (*p* < 0.001).

In the CA3c, no significant main effect of Surgery was observed (*p* = 0.802), but strong main effects of Age [F(2,144) = 16.722, *p* < 0.001] and Surgery × Age interaction [F(2,144) = 69.634, *p* < 0.001] were found. Post hoc tests revealed that 14- and 18-month-old sham rats exhibited lower pAkt1 expression compared to 3-month-old sham rats (*p* < 0.001) (Figure 3C). Pinealectomy significantly increased pAkt1 expression in 14-month-old rats relative to their respective sham controls (*p* < 0.001) whereas decreased it in 3-month-old rats compared to their matched sham controls (*p* < 0.001).

Similarly, in the CA3b, no main effect of Surgery was detected (*p* = 0.494), but a main effect of Age [F(2,165) = 10.290, *p* < 0.001) and Surgery × Age interaction [F(2,165) = 45.237, *p* < 0.001] were observed. Post hoc test showed reduced pAkt1 expression in 14- and 18-month-old sham rats compared to 3-month-old sham rats (*p* < 0.001) (Figure 3D). Pinealectomy significantly increased pAkt1 expression in 14-month-old rats compared to their matched sham controls (*p* < 0.001) and decreased it in 3-month-old rats relative to their matched controls (*p* < 0.001).

For pAkt1 expression in the CA3a region, two-way ANOVA revealed significant effects of Age [F(2,134) = 13.821, *p* < 0.001] and Surgery × Age interaction [F(2,134) = 18.764, *p* < 0.001], while no significant main effect of Surgery was found [F(1,134) = 1.812, *p* = 0.181]. Post hoc analyses indicated an age-related decline in pAkt expression within sham-operated groups: pAkt1 levels were significantly lower in 14- and 18-month-old rats compared to 3-month-old sham rats (*p* < 0.001 for both comparisons) (Figure 3E). Pinealectomy induced a significant reduction in pAkt1 expression in young adult (3-month-old) rats (*p* < 0.001), as evidenced by lower pAkt1 levels in pinealectomized rats compared to age-matched sham-operated controls.

Two-way ANOVA revealed significant main effects of Surgery [F(1,282) = 67.827, *p* < 0.001], Age [F(2,282) = 51.932, *p* < 0.001], and a significant Surgery × Age interaction [F(2,282) = 40.949, *p* < 0.001] in the CA1 region. Post hoc analyses indicated an age-related decline in pAkt1 expression within the sham-operated groups: pAkt1 expression was significantly lower in 14- and 18-month-old rats compared to 3-month-old sham rats (*p* < 0.001 for both comparisons) and 18-month-old rats also had significantly lower levels than 14-month-old sham rats (*p* < 0.001) (Figure 3F). Furthermore, pAkt1 expression was significantly reduced by pinealectomy in 3-month-old rats compared to age-matched sham-operated controls (*p* < 0.001).

Overall, these findings demonstrate that pAkt1 expression in the dHipp is highest in young adult sham animals, localized in neurons, and gradually declines with age under normal physiological conditions (Figure 1, Figure 2 and Figure 3). Pinealectomy disrupts this trajectory by reducing pAkt1 levels at 3 months but triggering a compensatory increase at 14, particularly in the DG, CA3c and CAb, as confirmed by both immunohistochemical (Figure 1) and quantitative analyses (Figure 3).

### 2.2. Effects of Pinealectomy and Aging on NF-kB Expression in Different Hippocampal Regions

Pinealectomy significantly altered NF-kB protein levels across various hippocampal subregions, with both Age and the interaction between Age and Surgery contributing to these changes in several areas.

Immunohistochemical detection of NF-κB in the dHipp revealed distinct regional and age-dependent patterns of cellular immunoreactivity (Figure 4). In the MoDG, only sparse NF-κB-positive cells were observed, with no apparent differences between groups.

In the GrDG, 3-month-old sham-operated rats exhibited low numbers of weakly stained NF-κB-positive cells, primarily characterized by a diffuse cytoplasmic signal (Figure 4(A2)). Aging resulted in a moderate increase in immunopositive profiles in sham animals, particularly at 14 months (Figure 4(C2)) and 18 months (Figure 4(E2)). A marked increase in the number and intensity of labeled cells was observed in 14-month-old pinealectomized animals (Figure 4(D2)), where numerous cells displayed intense nuclear and cytoplasmic staining, in sharp contrast to age-matched controls. This nuclear localization suggests transcriptional activation of NF-κB in cells within the granule layer.

In the CA3c subregion, immunoreactive cells were more numerous in all pinealectomized groups compared to their respective controls, with the most significant increase again detected at 14 months (Figure 4(D3)). Labeled cells exhibited dense cytoplasmic staining, and several showed a nuclear signal as well. Aged sham animals also showed slightly elevated numbers of positive cells compared to young controls, although at much lower levels than their pinealectomized counterparts.

In CA3b, NF-κB-positive cells were consistently scarce across all experimental groups. In CA3a, aging was associated with a modest increase in immunoreactive cells in sham animals, while pinealectomy led to a clear elevation at 3 months. Staining in this region was predominantly cytoplasmic, with occasional nuclear localization.

In CA1, increased numbers of NF-κB-positive cells were observed in 3-month-old pinealectomized rats compared to sham controls, with labeling localized to the soma and, in some cases, to the nuclei. No notable differences were observed in older groups.

Taken together, the immunohistochemical findings indicate that pinealectomy leads to a region-specific upregulation of NF-κB expression, most pronounced in the GrDG and CA3c at 14 months of age. The detection of nuclear immunoreactivity in these regions supports the activation of NF-κB signaling and suggests enhanced pro-inflammatory activity in response to melatonin deficiency during middle age.

To identify the primary cellular compartment of NF-κB under baseline conditions, we performed triple-label immunofluorescence staining in the CA1 region. We combined Hoechst staining with NF-κB staining and one of three cell-type markers: NeuN, GFAP, and Iba1 (Figure 5). Within the pyramidal cell layer (stratum pyramidale), we observed prominent NF-κB immunoreactivity in NeuN-positive pyramidal neurons. These neurons exhibited cytoplasmic and perinuclear labeling, along with distinct intranuclear puncta. Merged images revealed strong colocalization of NeuN and NF-κB.

In contrast, NF-κB colocalization in glial cells was limited. GFAP-positive astrocytes showed sparse but reproducible overlap with NF-κB, which was typically located in short segments of proximal processes near perisomatic or juxtasomatic sites (Figure 5). There, astrocytic processes align along the border of the stratum pyramidale and stratum radiatum. Outside these segments, the GFAP and NF-κB channels were largely separate. Because GFAP does not label astrocytic nuclei, we cannot draw conclusions about the presence of NF-κB in these nuclei based on this data. Iba1-positive microglia displayed minimal overlap with NF-κB and exhibited no consistent pattern within the stratum pyramidale.

Both Surgery and Age did not affect NF-kB expression in the MoDG (*p* > 0.05) (Figure 6A). In the GrDG, a main effect of Age [F(2,199) = 7.664, *p* < 0.001], Pinealectomy [F(1,199) = 15.093, *p* < 0.001] as well as Age x Pinealectomy interaction [F(2,199] = 19.943, *p* < 0.001] was found. Post hoc analysis indicated that 3-month-old sham rats had lower expression of NF-kB than 14- and 18-month-old sham rats (*p* < 0.01 and *p* < 0.001, respectively) (Figure 6B). Pinealectomy significantly increased NF-kB levels in 14-month-old rats (*p* < 0.001) compared to their sham-operated controls (Figure 6B).

In the CA3c, a main effect of Surgery [F(1,120) = 33.941, *p* < 0.001] and Surgery × Age interaction [F(2,120) = 5.134, *p* = 0.007] was detected, whereas the main effect of Age was not significant [F(2,120) = 2.233, *p* = 0.112]. Post hoc tests revealed that old sham rats exhibited increased NF-kB expression compared to the young adult rats (*p* < 0.05) while pinealectomy significantly increased NF-kB expression in 3-month-old (*p* < 0.01), and 14-month-old (*p* < 0.001 compared to their respective sham controls, respectively (Figure 6C). In the CA3b, no significant main effects of Surgery or Age were observed (*p* > 0.05) (Figure 6D). In the CA3a, a main effect of Surgery [F(1,128) = 5.794, *p* = 0.018] and Age [F(2,128) = 13.568, *p* < 0.001] as well as a significant Surgery × Age interaction (F(2,128) = 6.493, *p* = 0.002) was detected. Post hoc analysis revealed that 18-month old sham rats had higher NF-kB expression than the 3-month-old rats (*p* < 0.01) while pinealectomy significantly increased NF-kB expression in 3-month-old rats (*p* < 0.001) compared to sham controls (Figure 6E). In the CA1, main effects of Surgery [F(2,228) = 11.422, *p* < 0.001] and Age [F(2,228) = 14.987, *p* < 0.001], along with a significant Surgery × Age Interaction [F(2,228) = 8.405, *p* < 0.001], were found. Post hoc analysis showed that pinealectomy significantly increased NF-kB levels in 3-month-old rats (*p* < 0.001) compared to their sham-operated counterparts (Figure 6F).

Taken together, the immunohistochemical findings indicate that pinealectomy leads to a region-specific upregulation of NF-κB expression, most pronounced in the GrDG and CA3c at 14 months of age and CA3c, CA3a and CA1 at 3 months of age. The detection of nuclear immunoreactivity in these regions supports the activation of NF-κB signaling and suggests enhanced pro-inflammatory activity in response to melatonin deficiency in young adult and middle-aged groups.

### 2.3. Effects of Pinealectomy and Aging on Microglia Activation in Different Hippocampal Regions

In young adult sham-operated rats (3 months old), microglial cells across all analyzed hippocampal subregions (DG layers, CA3a-c, and CA1; Figure 7(A2–A6)) displayed morphological characteristics typical of a resting state. These cells exhibited small somata and thin, finely ramified processes, with minimal Iba1 immunoreactivity, indicating a low baseline neuroinflammatory status at this developmental stage.

Pinealectomy at the age of 3 months triggered pronounced microglial activation throughout the hippocampus (Figure 7(B2–B6)). Activated microglia displayed hypertrophic somata, thicker and shorter processes, and frequently formed clusters, strongly suggesting a robust neuroinflammatory response due to pineal gland removal at this early age.

At middle age (14-month-old), sham-operated rats demonstrated moderate baseline microglial activation across the hippocampal subregions (Figure 7(C2–C6)). Microglial cells appeared slightly more numerous and exhibited mild morphological hypertrophy, reflecting age-related neuroinflammatory changes. Nonetheless, the overall distribution and morphology remained relatively homogeneous without severe activation patterns.

The pinealectomy at 14 months displayed morphological had characteristics of a resting state within most of the hippocampal regions (Figure 7(D1–D6)).

In advanced-aged sham-operated rats (18-month-old, Figure 7(E2–E6)), microglia displayed intermediate activation profiles compared to younger sham animals, characterized by moderate morphological hypertrophy, slightly thicker and shorter processes, and a mild increase in cell density. These morphological traits reflect a baseline age-dependent inflammatory status inherent to normal aging.

However, pinealectomy also potentiated age-related neuroinflammation in the hippocampus of 18-month-old rats (Figure 7(F2–F6)). There was a dramatic increase in microglial density and a severe morphological transformation to an activated phenotype, especially within CA3 (CA3c and CA3b) regions. Microglial cells in these areas exhibited pronounced hypertrophy, amoeboid shapes, significantly enlarged somata, dramatically shortened processes, and extensive clustering, indicative of a chronic and robust neuroinflammatory state triggered by pinealectomy in older animals.

In the MoDG, two-way ANOVA revealed Surgery x Age interaction [F(2,76) = 11.982, *p* < 0.001]. Post hoc analyses showed that pinealectomy affected microglia activation by decreasing the transformation index (TI) in 3-month-old rats compared to age-matched sham controls (*p* = 0.021) (Figure 8A). In contrast, melatonin deficiency induced in the middle-aged rats caused resistance to microglial activation (higher TI in pin rats compared to matched sham rats) (*p* < 0.001). In the GrDG, two-way ANOVA showed main effects of Surgery [F(1,85) = 2.598, *p* = 0.111]. Post hoc comparisons revealed that pinealectomy significantly increased microglia activation by decreasing (TI) for Iba1 in 3-month-old rats compared to sham-operated controls (*p* = 0.011) (Figure 8B).

In the CA3c, two-way ANOVA revealed main effect of Surgery [F(1,265) = 4.096, *p* = 0.044], and Surgery × Age interaction [F(2,265) = 11.077, *p* < 0.001]. Pinealectomy significantly decreased TI for Iba1 in both 3- and 18-month-old rats relative to sham-operated controls (*p* = 0.003 for both comparisons), suggesting activated microglia due to pinealectomy (Figure 8C).

In the CA3b, significant effects were observed for Surgery [F(1,265) = 13.576, *p* < 0.001], and the Age [F(2,265) = 11394, *p* < 0.001]. Post hoc analyses indicated significantly activated microglia in 18-month-old sham rats compared to 3-month-old sham rats (*p* = 0.017), while pinealectomy significantly decreased TI for Iba1 in 3- and 18-month-old rats compared to age-matched sham-operated controls (*p* = 0.032 and *p* = 0.016), respectively (Figure 8D).

In the CA3a, two-way ANOVA showed main effects of Surgery [F(1,265) = 20.168, *p* < 0.001], and a significant Surgery × Age interaction [F(2,265) = 9.039, *p* < 0.001]. Post hoc analyses demonstrated that pinealectomy significantly decreased TI for Iba1 in 3-month-old rats compared to age-matched sham-operated controls (*p* = 0.006) (Figure 8E). In contrast, TI for Iba1 was increased in 14-month-old rats compared to their sham controls (*p* = 0.003).

In the CA1, a significant main effect of Surgery [F(1,2265) = 5. 515, *p* = 0.02] as well as Surgery x Age interaction was detected [F(2,265) = 14, 899, *p* < 0.001]. Pinealectomy caused activation of microglia (reduced TI of Iba1) in young adult rats compared to their sham controls (*p* = 0.0108) (Figure 8F). In contrast, as for the CA3c subfield, pinealectomy in the 14-month pin group made them resistant to activation of microglia compared with sham controls (*p* < 0.001).

Collectively, these findings suggest a clear interaction between aging and pinealectomy-induced microglial activation, with the most pronounced neuroinflammatory responses observed in young adult pinealectomized animals affecting all hippocampal subregions.

### 2.4. Effects of Pinealectomy and Aging on GFAP Expression in Different Hippocampal Regions

Immunohistochemical analysis of glial fibrillary acidic protein (GFAP) expression in the dHipp revealed age-dependent and pinealectomy-induced alterations in astrocytic morphology and activation (Figure 9).

At 3 months of age, sham-operated animals (Figure 9(A1–A6)) displayed mild to moderate astrocytic activation characterized by slender, moderately branched GFAP-positive astrocytic processes within the molecular layer of the MoDG, GrDG, and PoDG. Astrocytes in hippocampal subfields CA3c (Figure 9), CA3b, and CA3a exhibited uniformly low GFAP immunoreactivity, which was weakest in the CA1 region. Pinealectomy at this age resulted in slightly increased GFAP expression, particularly notable within the CA3c, CA3b and CA3a subfields, as evidenced by thicker astrocytic processes. Minimal changes were detected in CA1 compared to sham controls.

At 14 months of age, sham-operated animals also exhibited low astrocytic activation across the hippocampal regions, including the GrDG and PoDG layers, CA3c, CA3b, CA3a, and CA1 regions without noticeable increased GFAP immunoreactivity compared to their younger counterparts. Pinealectomy significantly intensified astrocytic activation, particularly within the GrDG, PoDG, and CA3c, as reflected by robust hypertrophy and increased GFAP expression.

At 18 months, sham animals exhibited prominently elevated GFAP immunoreactivity, indicative of naturally occurring age-related reactive astrogliosis. Astrocytes in the GrDG and PoDG demonstrated marked hypertrophy, characterized by densely packed, thickened processes. The CA3c and CA3b regions exhibited similar increases in astrocytic activation, characterized by conspicuously enhanced GFAP immunoreactivity, compared to younger animals. Pinealectomy further exacerbated astrogliosis, yielding the highest level of astrocytic hypertrophy and GFAP expression in the CA3b subregion relative to sham controls. CA1 also exhibited heightened astrocytic activation, which exceeded that of the corresponding sham-operated group

In the MoDG, two-way ANOVA revealed a significant main effect of Age [F(2,59) = 17.578, *p* < 0.001]. Post hoc comparisons showed that GFAP expression was significantly higher in 18-month-old sham-operated rats compared to 3-month-old rats (*p* < 0.001) as well as 14-month-old sham rats compared to 3-month-old rats (*p* < 0.001) (Figure 10A).

In the GrDG, no significant main effects of either Age (*p* > 0.05) or Surgery (*p* > 0.05) was detected (Figure 10B).

In the CA3c region, significant main effects of Surgery [F(1,59) = 29.107, *p* < 0.001], Age [F(2,59) = 0.842, *p* = 0.436] as well as Age × Surgery interaction [F(2,59) = 3.139, *p* = 0.05] was observed. Post hoc comparisons showed that GFAP expression was significantly higher in 18-month-old than 3-month-old sham rats (*p* = 0.015) and pinealectomy significantly increased GFAP expression in 3-month-old (*p* < 0.001) and 14-month-old rats (*p* = 0.011) relative to age-matched sham controls (Figure 10C).

In the CA3b, two-way ANOVA revealed significant main effects of Age [F(2,59) = 13.398, *p* < 0.001], Surgery [F(1,59) = 18.062, *p* < 0.001], and Surgery × Age interaction [F(2,59) = 14.495, *p* < 0.001]. GFAP levels in sham rats were significantly lower at 3 months compared to 18 months (*p* < 0.001) as well as 14 months compared to 18 months (*p* < 0.001). Pinealectomy significantly increased GFAP expression in both 3-month-old (*p* = 0.008) and 14-month-old rats (*p* < 0.001), compared to their respective sham controls (Figure 10D).

In the CA3a, significant main effects of Age [F(2,59) = 15.900, *p* < 0.001], Surgery [F(1,59) = 4.427, *p* = 0.04], and Surgery × Age interaction [F(2,59) = 11.741, *p* < 0.001] were observed. Post hoc analyses showed age-related increase in astrocyte activation (18-month-old sham rats vs. 14- and 3-month-old sham rats, *p* < 0.001; 14-month-old sham rats vs. 3-month-old sham rats, *p* = 0.004) (Figure 10E).

In the CA1, there was a significant main effect of Surgery [F(1,288) = 15.421, *p* < 0.001], Age effect [F(2,288) = 15.430, *p* < 0.001] as well as the Surgery × Age interaction [F(2,288) = 14.000, *p* < 0.001]. Post hoc analyses indicated that GFAP levels were significantly lower in 3-month-old sham rats compared to 18-month-old sham rats (*p* < 0.001). Additionally, pinealectomy significantly increased GFAP expression in 3-month-old rats relative to their matched controls (*p* < 0.001) (Figure 10F).

Collectively, these findings underscore a progressive, age-dependent increase in hippocampal astrocytic activation and hypertrophy, markedly potentiated by pinealectomy. These data support the hypothesis that melatonin, produced by the pineal gland, critically modulates age-associated neuroinflammatory responses in hippocampal regions integral to cognitive function.

## 3. Discussion

Overall, the results of the double immunofluorescence staining for pAkt1 suggest that basal Akt1 phosphorylation as well as NF-κB are primarily neuronal and restricted to pyramidal neurons under sham conditions at three days of age. While pAkt is absent in astrocytes and microglia, there is a limited overlap with astrocytes, which is confined to specific perisomatic areas for NF-kB. The latter also has little association with microglia. These results suggest that the Akt and NF-κB signaling pathways predominantly exhibit neuronal activity at baseline. Additionally, astrocytes modestly contribute to NF-kB distribution in specific regions.

Pinealectomy performed on 3-month-old rats resulted in the highest level of inflammation in the dHipp (Table 1). In contrast, melatonin deficiency had a partial effect on 14- and 18-month-old rats in neurons, with responses that varied based on specific areas of the dHipp. While the inflammatory markers Akt and NFkB in neurons were unaffected in the oldest rats, melatonin deficiency in the 14-month-old rats conferred resilience to inflammation through Akt resistance in neurons and in microglia.

### 3.1. Aging and Pinealectomy in Young Adult Rats Reduce pAkt1 Expression in dHipp, While Melatonin Deficiency in Older Rats Increases It

The pAkt is involved in physiological pathway associated with proliferation, growth, metabolism, and survival [4]. This signaling molecule plays a crucial neuroprotective role in aging, yet it also has deleterious consequences on cognitive functions, abnormal cellular functions, and predisposes individuals to neurodegeneration. Most studies have demonstrated a tendency for a decline in pAkt1 levels with age. In old hepatocytes, the activity of the Akt pathway is reduced resulting in diminished survival [31]. Furthermore, reduced pAkt1 levels in the hippocampus with age have also been reported in many studies [4,32,33]. pAkt1 is the active form of the protein kinase Akt1, a crucial player in cellular signaling pathways. Its phosphorylation at key sites (the threonine residue (Thr308) and the serine residue (Ser473)) can led to complete activation and thereby maximize catalytic activity [34]. Diminished expression of pAkt1 with aging may be associated with impaired synaptic plasticity and, consequently, cognitive decline [35]. Aging reduces pAkt in the DG, which might correspond with reduced neurogenesis [36,37,38]. Reduced phosphorylation of AktSer473 in the hippocampus is reported in aging of senescence-accelerated mouse SAMP10 [39]. Therefore, disturbed activity of Akt signaling in the hippocampus may increase the risk of neurodegenerative processes (e.g., Alzheimer’s disease).

Our finding that the dHipp exhibits change in pAkt1 expression with aging in the sham groups supports the hypothesis that this signaling molecule may be considered a suitable marker of aging.

Literature data are in line with the suggestion that endogenous melatonin can trigger neuronal PI3K/Akt pathway [40,41]. It is reported that melatonin supplementation can enhance Akt signaling (Akt to pAkt) via activation of MT1 and MT2 melatonin receptors in different cell types, including neurons and thus, can have a protective effect in aging via this pathway [4,38,41,42]. Akt phosphorylation is a crucial activation step often linked to the neuroprotective effects of melatonin, particularly in circumstances such as strokes where cell damage occurs [43,44]. Aging predisposes melatonin deficiency due to a sharp decline in melatonin production and can alleviate pAkt. Melatonin production is likely to reduce pAkt1 signaling in the hippocampus, although direct studies specifically measuring this effect are somewhat limited. In the present study, we reported that melatonin deficiency related to the removal of the pineal gland could affect Akt signaling in an age-dependent manner. Curiously, while young adult rats with pinealectomy demonstrated lower expression of pAkt1 in the DG and the cornus ammonis, similar to aging sham rats, the middle-aged rats exhibited increased expression of this signaling molecule compared to their matched sham group at the same age. Moreover, 18-month-old rats with pinealectomy also showed a tendency to elevation of pAkt1. We can assume that pinealectomy leads to the higher vulnerability to Akt pathway, which is involved in neuroprotection, while melatonin deficiency in the middle age and partly in old age makes the hippocampus more resistant to impairment of this signaling system.

### 3.2. Pinelectomy Induced Age- and Structure-Dependent Increased Expression of NF-κB in the dHipp

One of the most densely distributed form of NF-κB in the hippocampus is the p65/p50 heterodimer, suggesting it plays a pivotal role in synaptic plasticity and cognitive processes that control the function of dendrites [45]. The overactivated transcription factor NF-κB in aging is involved in processes closely related to neurodegeneration, such as neuroinflammation, oxidative stress, impaired neurogenesis causing cognitive decline [3,46]. This key transcription factor demonstrated higher activation in response to cellular stress in various tissues. The detected upregulation of NF-kB, both in physiological conditions such as aging and in a mouse model deficient in a gene critical for DNA repair, suggests a similar mechanism underlying normal aging and pathogenesis associated with DNA damage accumulation [7]. These experimental findings support the hypothesis that there is a bidirectional link between inflammation and the aging process. In the aged hippocampus, NF-kB plays a role in both neuroinflammation and age-related neurodegeneration. Studies suggest that NF-κB activation increases with age in several tissues, including the brain, and this increase is associated with enhanced inflammation and impaired cognitive function [47]. Our findings align with the literature, which reveals lower NF-kB expression in young adult controls compared to their matched older groups in the GrDG, CA3c and CA3a subfields of the dHipp.

In physiological conditions, melatonin can suppress NF-κB activation in different cell types, including microglia and neurons in the hippocampus, through scavenging free radicals and thus, attenuating oxidative stress, suppressing kinases that trigger phosphorylation of IkB, and 3) enhancing SIRT1 signaling pathway, which can negatively regulate NF-κB [48,49,50,51,52].

The reducing effect of melatonin on NF-κB activity in the hippocampus is multifaceted, involving both direct scavenging of free radicals and receptor-mediated signaling pathways [48,53]. Melatonin can also help to reduce neurodegeneration and neuronal loss in the hippocampus, potentially by modulating signaling pathways involved in neuroinflammation, such as the TLR4/MyD88/NF-κB pathway. In contrast, we can speculate that melatonin deficiency may stimulate NF-κB signaling in the hippocampus through a logical interrelated sequence of steps involving the oxidative stress-induced release of pro-inflammatory cytokines (TNF-α, IL-1β, IL-6), as well as fostering NF-κB nuclear translocation due to higher pIkB. In this regard, we recently reported that melatonin deficiency induced by the removal of the pineal gland is associated with increased oxidative stress in the hippocampus of young adult rats [26].

Previous studies have reported that the Akt signaling pathway regulates the expression of Hsp70, a mechanism closely related to the function of Hsp90 [54]. Hsp is also crucial for modulating the NF-kB signaling pathway to enhance oxidative stress tolerance [55]. There is a positive correlation between oxidative stress resistance and the activation of ERK1/2 or Akt signaling pathways [31]. Melatonin, through its role as a direct catcher of ROS, as well as by activating antioxidant enzymes, can effectively protect various cell types from oxidative stress and further inflammatory damage [56]. Moreover, H_2_O_2_-induced modulation of extracellular signaling molecules (ERK and Akt) and some of their regulatory protein effectors (Hsp and NF-kB) is also under the direct control of melatonin. Although explicit measurements of NF-κB in the hippocampus of pinealectomized animals are limited, our findings reveal, for the first time, that the increasing effect of pinealectomy on NF-kB expression is age-dependent and differs between the granular dentate gyrus and the Cornu Ammonis of the dHipp.

These findings agree with our previous reports, which revealed that melatonin deficiency induced by pinealectomy suppresses MT_1_/MT_2_ receptor–ERK/CREB signaling pathways [57,58]. This is known to be negatively affected by NF-κB-associated inflammation [59,60]. Interestingly, we have recently reported that Hsp70 in the FC was reduced only in the middle-aged rats with pinealectomy, but was not affected in young adult and old rats compared to their respective sham groups [27].

### 3.3. Pinealectomy Increased the Activation of Microglia in the Hippocampus of Young Adult and Aged Rats, but Made Middle-Aged Rats Resistant to Activation

In the aging hippocampus, studies have observed alterations in Iba1 expression, which may reflect changes in microglial activation and density. Thus, aged KK-Ay/TaJcl mice (a model of type 2 diabetes) exhibited higher hippocampal levels of Iba1 than younger mice, concomitant with behavioral impairments, including those in motor activity and memory [61]. Furthermore, the pro-inflammatory response of microglia was also associated with higher expression of other markers, such as CCR7 and inducible nitric oxide synthase (iNOS), as well as tumor necrosis factor alpha. Interestingly, while early-aged gerbils exhibited reduced Iba1 immunoreactivity, older animals had overactive microglial cells [62]. Unlike the reported age-related increase in microglial activation in the dHipp, in the present study, older sham rats exhibited changes in CAb region-specific microglial activation. This discrepancy may be explained by strain- and model-specific differences in the state of microglia. However, our results revealed that pinealectomy is directly associated with alterations in microglial activity in the hippocampus, marked by changes in the TI of the Iba1 in the youngest and oldest rats. In contrast, the microglial activation status remains unchanged in middle-aged rats with pinealectomy. The observed increased vulnerability of the youngest group to neuroinflammation in the hippocampus due to melatonin deficiency corresponds to changes in Akt and NF-kB expression, as well as to our previous findings on cognitive decline and increased oxidative stress [26,57], suggesting that these alterations are crucial factors contributing to pinealectomy-related neuroinflammatory processes. Furthermore, whereas middle-aged rats were resistant to melatonin deficiency-induced microglia activation, the susceptibility to microglia-related neuroinflammation in old rats was similar to that of young adults. This result was unexpected, given the effects of these two groups on Akt and NF-kB. However, our previous study showed that cognitive decline in rats with pinealectomy in 14-month-old rats correlated with reduced BDNF/ERK1/2/CREB signaling in some regions of the hippocampus, while ERK 1/2 and its phosphorylated form were increased in the other areas compared to the matched sham group, suggesting that melatonin-associated changes in this age are complex. Considering Pierpaoli and Bulian’s (2005) hypothesis that pinealectomy can suppress the aging process in middle-aged mice, as demonstrated by some aging markers in plasma [30], our findings in rats with induced melatonin deficiency at the same age indicate that this age period is crucial for the aging process, especially regarding the role of the melatonin system. It is possible that there are partial adaptive mechanisms that compensate for the lack of hormonal function through unknown pathways.

However, our recent and current results demonstrate that the removal of the pineal gland may also have a harmful effect on middle-aged rats, leading to an exacerbation of the aging process, suggesting a complex role of the hormone across the lifespan that is still not fully explored and requires further analysis. Pinealectomy-induced susceptibility to cognitive changes in 3- and 14-month-old rats was associated with decreased expression of molecules involved in plasticity changes in the hippocampus, closely associated with memory processing, including the BDNF/ERK1/ 2/CREB pathway. We found that middle-aged rats with pinealectomy had increased expression of both ERK1/2 and its phosphorylated form in some regions of the hippocampus, including the GrDG and the CA3c and CA3b subfields, compared to the matched sham group.

Melatonin has been demonstrated to have potent anti-inflammatory activity, including ability to protect neurons in the hippocampus, particularly induced by stress [63,64]. Thus, the hormone can affect stress-induced changed morphology and reduces the expression of markers Iba1, which is indicative of microglial activation, but also co-localized Iba1 and NLRP3, which is another marker of neuroinflammation in the hippocampus [63,65,66].

### 3.4. Age- and Pinealectomy-Induced Vulnerability to Astrocytes Activation Is Region-Specific for Young Adult, Middle-Aged and Old Rats

Unlike, microglia, we have reported age-associated elevation of GFAP expression in a region-specific manner in the dHipp. The impact of age on astrocyte activation within the hippocampus has been demonstrated to be of significance [67,68]. This activation has been shown to affect cognitive function and the process of neurodegeneration, which is characterized by the progressive loss of neurons in the brain [67]. As organisms age, astrocytes tend towards reactivity; this is evidenced by increased expression of certain proteins (e.g., GFAP) [68]. This reactive astrogliosis has been observed to be associated with a range of factors, including neuroinflammation, altered neurotransmitter regulation, and a decline in neuroplasticity. It is possible that these factors may contribute to cognitive deficits observed during the aging process [57]. However, our findings suggest that neuroinflammation associated with astroglial reactivity is region-specific in the dHipp. Moreover, young adult and middle-aged groups with pinealectomy were affected and have increased expression of activated astrocytes in selected areas of the dHipp. This result was expected and confirmed with previous reports suggesting the pivotal role of melatonin on astrocyte functions. The beneficial impact of the hormone on astrogliosis was reported in two different animal models of Alzheimer diseases [69,70]. Thus, having in mind that melatonin has antioxidant properties that help regulate astrocyte function it can be speculated that pinealectomy diminishes these protective effects, potentially leading to increased vulnerability of neural tissue to damage. Reduced melatonin levels, may elevate oxidative stress and pro-inflammatory cytokines, contributing to astrocyte activation or reactive gliosis associated with neuroinflammatory responses. This is often characterized by increased GFAP expression and morphological changes.

## 4. Materials and Methods

### 4.1. The Animals

Male Wistar rats were used in this study and obtained from the vivarium of the Institute of Neurobiology, BAS. Animals were divided into three age groups: 3-, 14-, and 18 months. Their body weights ranged from 250 to 600 g, depending on age. Rats were housed under standard laboratory conditions: a 12 h light/dark cycle (lights on at 07:00 a.m.), a stable ambient temperature of 21 °C, and relative humidity maintained at approximately 45%. Animals were housed in standard Plexiglas cages, grouped in threes or fours, with cage size adjusted to accommodate age and size. Food and water were available ad libitum, except during specific experimental procedures. The study included six experimental groups, categorized by age and surgical treatment (n = 5–6 per group): (1) 3-month-old sham-operated rats; (2) 3-month-old pinealectomized (pin) rats; (3) 14-month-old sham rats; (4) 14-month-old pin rats; (5) 18-month-old sham rats; (6) 18-month-old pin rats.

All experimental procedures were conducted in accordance with the European Communities Council Directive 2010/63/EU. Animal protocols were reviewed and approved by the Bulgarian Food Safety Agency under project number #300/N°5888–0183.

### 4.2. Surgery

Animals in each age group were randomly assigned to either a sham-operated or pinealectomized subgroup. All surgical interventions were performed under general anesthesia with 2.5% isoflurane. Animals were positioned in a stereotaxic frame (Stoelting, Wood Dale, IL, USA), and pinealectomy was carried out following the procedure originally described by Hoffman and Reiter (1965) [71]. The surgical approach involved removal of the pineal gland via the cranial window, as previously adapted in our laboratory [72]. Sham-operated rats underwent the same surgical procedures, excluding the removal of the pineal gland.

### 4.3. Immunohistochemistry and Immunofluorescence

Euthanasia was conducted one month after the sham surgery or the pinealectomy procedure. Animals from all three age cohorts of Wistar rats (n = 5 per group) were administered deep anesthesia using urethane at a dosage of 1500 mg/kg. Subsequently, they underwent transcardial perfusion—initially with 150 mL of ice-cold 0.05 M phosphate-buffered saline (PBS), and then with 500 mL of chilled 4% paraformaldehyde (PFA) in 0.1 M phosphate buffer (PB), adjusted to pH 7.4. Once perfused, the brains were carefully extracted and immersed in the same fixative overnight at 4 °C for post-fixation. The brain tissue was then processed by embedding in paraffin and sectioned into 6 μm thick coronal slices.

Immunohistochemical detection of Akt (Phospho-Akt1), NF-kB, Iba1 (AIF1) and GFAP was carried out on paraffin-embedded tissue sections following deparaffinization. The staining procedure employed the UltraTek HRP Anti-Polyvalent Detection System (AFN600, ScyTek Laboratories, Logan, UT, USA). For antigen unmasking, sections were placed in a water bath (WB-4MS model) and heated to 95 °C for 20 min in 0.01 M citrate buffer (pH 6.0). Following this step, the slides were rinsed in TBST buffer (Tris-buffered saline containing 0.05% Tween-20, pH 7.6). Endogenous peroxidase activity was quenched by incubating the sections in 3% hydrogen peroxide in distilled water for 10 min at room temperature. To reduce background staining, non-specific binding was blocked using Super Block solution (ScyTek Laboratories, Logan, UT, USA) for 10 min. Afterward, the slides underwent three brief washes in TBST, followed by biotin blocking treatment (Cat. No. BBK120) using reagents provided by the same manufacturer. The sections were incubated according to the kit instructions: 15 min with part A, washing, and 15 min with part B. All antibody solutions were prepared using ScyTek’s Tris-based primary antibody diluent (ATG125). The sections were incubated overnight at 4 °C with the following primary antibodies: rabbit polyclonal anti-Phospho-AKT1 (Thr308) (1:200, Affinity Biosciences, AF0832), rabbit polyclonal anti-NF kappaB p100/p52 (1:400, Affinity Biosciences, Cincinnati, OH, USA, AF6373), mouse monoclonal anti-AIF1 (1:500, Elabscience, Houston, TX, USA, E-AB-70373) and mouse monoclonal anti-GFAP (1:1000, Elabscience, Houston, TX, USA, E-AB-70205). On the next day, the slides were treated sequentially with UltraTek Biotinylated Secondary Reagent and HRP-conjugated detection reagent (Cat. No. AFN600, ScyTek Laboratories, Logan, UT, USA). Visualization was achieved using the DAB Chromogen/Substrate Kit (ScyTek Laboratories, Logan, UT, USA). Finally, the tissue sections were counterstained with hematoxylin, dehydrated through graded alcohols and coverslipped. For controls, the primary antibody was replaced with antibody diluent.

Fluorescent detection used a similar protocol starting from the Superblock pretreatment (Waltham, MA, USA). The primary antibodies for immunofluorescence were mouse monoclonal [1B7] antibody to NeuN (1:3000, Antibodies.com, St. Louis, MO, USA, A85405), mouse monoclonal anti-GFAP antibody (1:400, Elabscience, E-AB-70205), mouse monoclonal anti-AIF1 (Iba1) antibody (1:400, Elabscience, E-AB-70353), rabbit polyclonal anti-Phospho-AKT1 (Thr308) (1:200, Affinity Biosciences, Cincinnati, OH, USA, AF0832) and rabbit polyclonal anti-NF kappaB p100/p52 (1:400, Affinity Biosciences, AF6373). A lipophilic fluorescent stain Hoechst 33342 Ultra Pure grade (CAS 23491-52-3) (1:1000, Santa Cruz Biotechnology, Inc., Dallas, TX, USA) for DNA labeling was used. Secondary antibodies goat anti-rabbit IgG (H+L)(AF488 conjugated) (1:100, Elabscience, E-AB-1055) and goat anti-mouse IgG(H+L) (AF594 conjugated) (1:100, Elabscience^®^, E-AB-1059) were used for 1 h at room temperature in the dark. The slides were mounted with FluoreGuard Mounting Medium (Hard Set) (FMH-060, ScyTek Laboratories, Inc., Logan, UT, USA) and observed under a fluorescent microscope Leica TCS SPE equipped with a Leica Application Suite X (LAS X) Microscope Software, Version 3.5.7.23225 (Leica Microsystems GmbH, Wetzlar, Germany).

#### 4.3.1. Photodocumentation and Image Analysis

After immunohistochemical reactions for visualizing Akt, NF-kB, Iba1 and GFAP in certain brain structures, the slides were examined and photographed with a Leica DM1000 bright-field microscope equipped with a Leica DFC 290 digital imaging system. The light source and camera settings were kept the same for every single image. The images were recorded in TIF format and processed for removal of artifacts using Adobe Photoshop CS5 software (version 12.x; Adobe Systems Inc., San Jose, CA, USA). To quantify cell density, the stained sections were digitalized to produce images of the respective sections and regions of interest (ROIs). These images were analyzed using the ImageJ analysis program (National Institutes of Health, Bethesda, MD, USA). Immunopositive cell counting was performed using the Cell Counter plugin, available on the microscopy server within ImageJ using the “Cell Counter” plugin, version 2.22 (K. De Vos). The ROI area was manually outlined, and the cells of interest were counted by an experienced researcher blinded to the treatment groups. Analyses were performed on 6 µm-thick coronal paraffin sections of the hippocampus at −3.12 mm to −4.68 mm Bregma, according to the rat brain stereotaxic atlas [57].

#### 4.3.2. Transformation Index

Brightfield micrographs of 6 μm paraformaldehyde-fixed, paraffin-embedded brain sections immunostained for Iba1 with DAB chromogen and hematoxylin counterstain were acquired on a research microscope, Leica DM1000, equipped with a digital camera, Leica DFC 290, and images were saved as TIFF with preserved metadata. Predefined hippocampal subfields (e.g., MoDG, GrDG, CA3a–c, CA1) were sampled as non-overlapping fields using a systematic uniform random scheme while avoiding folds, precipitate, and large vessels. Image processing was performed using the ImageJ analysis program (National Institutes of Health, Bethesda, MD, USA) with H-DAB color deconvolution to isolate the DAB channel, background subtraction, and application of a single pre-registered global threshold (e.g., Default) held constant within each analysis. Inclusion required individual, non-overlapping Iba1-positive microglia fully contained within the field of view, with clear soma and discernible processes; exclusions comprised border-truncated cells, aggregates not reliably separable, and strong artifacts. For each segmented cell, area (A, μm^2^) and perimeter (*p*, μm) were extracted, and the transformation index computed as the inverse circularity TI(inv) = P^2^/(4πA), which increases with ramification and decreases with compact (amoeboid) morphology.

#### 4.3.3. GFAP-Positive Area

GFAP-positive area fraction (%GFAP). Astroglial immunoreactivity was assessed on paraffin sections processed for GFAP with DAB chromogen and hematoxylin counterstain. Brightfield micrographs were acquired at fixed magnification and exposure and saved as TIFF. Images were analyzed in ImageJ after H–DAB color deconvolution to isolate the DAB channel. Background was subtracted, a single batch-specific global threshold was applied, and minimal morphological filtering was used to suppress speckle without eroding fine processes. A binary tissue mask excluding slide background and artifacts (folds, precipitate, large vessels) defined the reference area. %GFAP was computed from calibrated areas (µm^2^) as 100 × GFAP + positive area/tissue area. Non-overlapping fields were sampled using a systematic uniform random scheme within predefined hippocampal subfields. All analyses were performed blinded, with visual quality control of mask overlays before data export.

### 4.4. Statistical Analysis

Results were assessed by a two-way ANOVA (factors: Age and Surgery). In case of significance, a Bonferroni post hoc test was applied. Data are presented as mean ± S.E.M. For statistical analyses and preparation of figures SigmaStat^®^ 11.0 and GraphPad Prism 6 software were used. As a criterion for significant differences *p* ≤ 0.05 was accepted.

## 5. Limitation and Future Direction

A limitation of this study is that we have primarily focused on the impact of pinealectomy on neuroinflammation rather than the melatonin deficiency in the whole organism. Studying melatonin deficiency exclusively through pinealectomy (the removal of the pineal gland) provides useful insights, but it also introduces several limitations that can affect the broad applicability of findings—especially when examining complex inflammatory signaling pathways, such as Akt, NF-κB, microglial, and astroglial responses. The surgical procedure of removing the pineal gland results in a deficit that is primarily confined to the melatonin contained in the blood and its associated hormonal function. However, beyond its classical production by the pineal gland, melatonin is also synthesized locally in the brain, including by neurons, astrocytes, and microglia as an important signaling molecule [73,74,75] suggesting its implications for neuroprotection and neuroinflammation as autocrine or paracrine hormone. Moreover, the latter local source of melatonin is supposed to partially compensate for hormonal dysfunction, leading to an underestimation of the inflammatory impact of complete melatonin loss. Therefore, to gain a more in-depth and precise understanding of the impact of melatonin deficiency in the hippocampus, future studies are needed that combine pinealectomy with pharmacological inhibitors or melatonin receptor knockouts to differentiate the effects of blood or non-pineal melatonin sources on neuroinflammation in the hippocampus. Nevertheless, by using pinealectomy as a tool of studying the role of deficiency of the primary source of circulating melatonin, it allows examining how the loss of melatonin as a hormone affects hippocampal pathways such as Akt and NF-κB as well as how melatonin loss contributes to glial reactivity (e.g., microglial activation and astrogliosis), which are central features of hippocampal neuroinflammation. In addition, considering that melatonin is a key regulator of circadian rhythms, this information will contribute to understanding the effect of circadian rhythm disruption as a neuroinflammatory trigger.

## 6. Conclusions

Whilst the protective impact of melatonin functioning as a hormone on neuroinflammation has been demonstrated and confirmed in different animal models, the present study is the first to report that removal of the pineal gland, which induces melatonin deficiency, exerts an age-dependent detrimental effect on different signaling pathways related to inflammatory responses in neurons, as well as glia in region-specific manner in the dHipp. Our findings confirm that the expression of pAkt, NFκB, Iba1, and GFAP within the hippocampus is not uniform, because the hippocampus is both anatomically and functionally heterogeneous. Variability across subfields of cornu ammonis and DG can reflect differences in cell type composition, baseline activity, as well as vulnerability to melatonin deficiency.

It is evident that young adult rats appear to be particularly susceptible to inflammation in the hippocampus due to a deficiency of melatonin in the blood. The middle-aged phase of life appears to be a pivotal juncture, wherein alternative, as yet unidentified, mechanisms emerge, potentially counteracting the heightened vulnerability to inflammation associated with melatonin deficiency. Consequently, this age group exhibits a degree of resilience to such inflammatory influences.

## Figures and Tables

**Figure 1 ijms-26-08093-f001:**
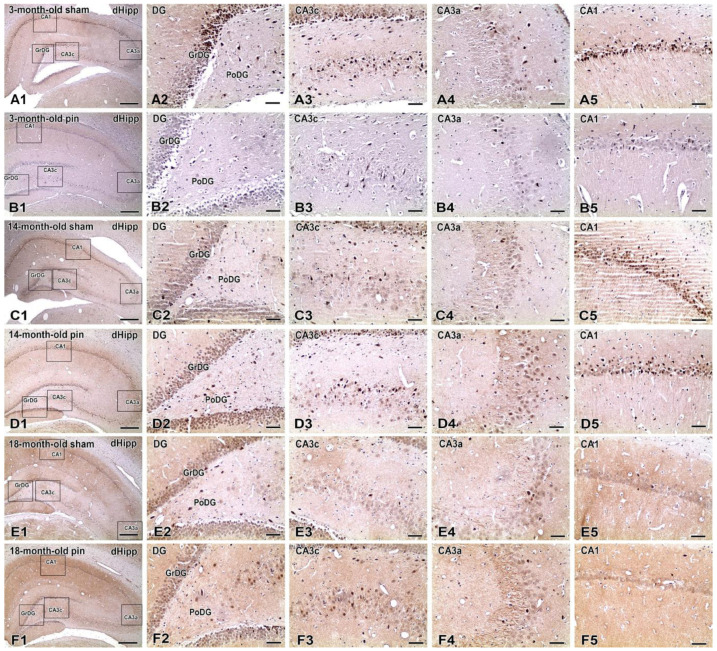
Immunohistochemical localization of pAkt1 in the dorsal hippocampus (dHipp) of sham-operated and pinealectomized male rats at different ages (3, 14, and 18 months). Representative low-magnification micrographs (**A1**–**F1**) show coronal sections through the dHipp, highlighting the analyzed subregions: dentate gyrus (DG), including the molecular (MoDG), granular (GrDG), and polymorphic (PoDG) layers, as well as CA3c, CA3a, and CA1 subfields of the Ammon’s horn. High-magnification micrographs (**A2**–**A5** through **F2**–**F5**) illustrate pAkt1 immunoreactivity in these regions. In 3-month-old sham-operated rats (**A1**–**A5**), strong pAkt immunoreactivity is visible, particularly in CA1, CA3c, and all layers of the DG. In age-matched pinealectomized animals (**B1**–**B5**), a clear reduction in pAkt1-positive cell density is observed across all subregions, especially in CA1 and CA3a. At 14 months, sham animals (**C1**–**C5**) exhibit moderate pAkt1 expression in the same regions, while pinealectomized rats (**D1**–**D5**) show markedly higher pAkt1 labeling, most notably in CA3c and the DG, consistent with pinealectomy-induced upregulation. In 18-month-old sham animals (**E1**–**E5**), pAkt1 expression remains moderate in the CA regions and the DG, whereas pinealectomized rats (**F1**–**F5**) display widespread and intense immunoreactivity, particularly in the CA3c and MoDG. Scale bars = 500 µm (**A1**,**B1**,**C1**,**D1**,**E1**,**F1**), 100 μm (**A2**–**A5**,**B2**–**B5**,**C2**–**C5**,**D2**–**D5**,**E2**–**E5**,**F2**–**F5**).

**Figure 2 ijms-26-08093-f002:**
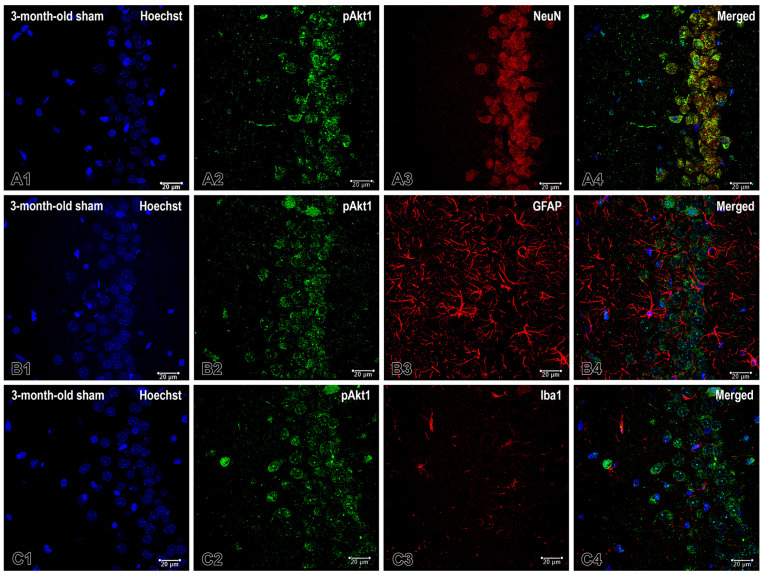
The cellular distribution of pAkt1 in the cornu ammonis area 1 (CA1) of the dHipp was examined in 3-month-old rats that underwent a sham operation. Representative confocal micrographs depict pAkt1 (green) in neuronal and glial cells, with nuclei counterstained with Hoechst (blue). Double labeling with the neuronal marker NeuN (red) reveals intense pAkt1 immunoreactivity in NeuN-positive pyramidal neurons within the CA1 pyramidal cell layer. This produces prominent somatic/perisomatic labeling and extensive pAkt1-NeuN overlap in the merged image (**A1**–**A4**). (**B1**–**B4**) Co-staining with the astrocytic marker glial fibrillary acidic protein (GFAP; red) reveals an absence of the pAkt1 signal in GFAP-positive astrocytes across the stratum radiatum and stratum oriens. Merged images demonstrate no colocalization. (**C1**–**C4**) Co-staining with the microglial marker ionized calcium-binding adapter molecule 1 (Iba1; red) likewise reveals no pAkt1 signal in Iba1-positive microglia, with no co-localization in the merged channels. Scale bar: 20 μm.

**Figure 3 ijms-26-08093-f003:**
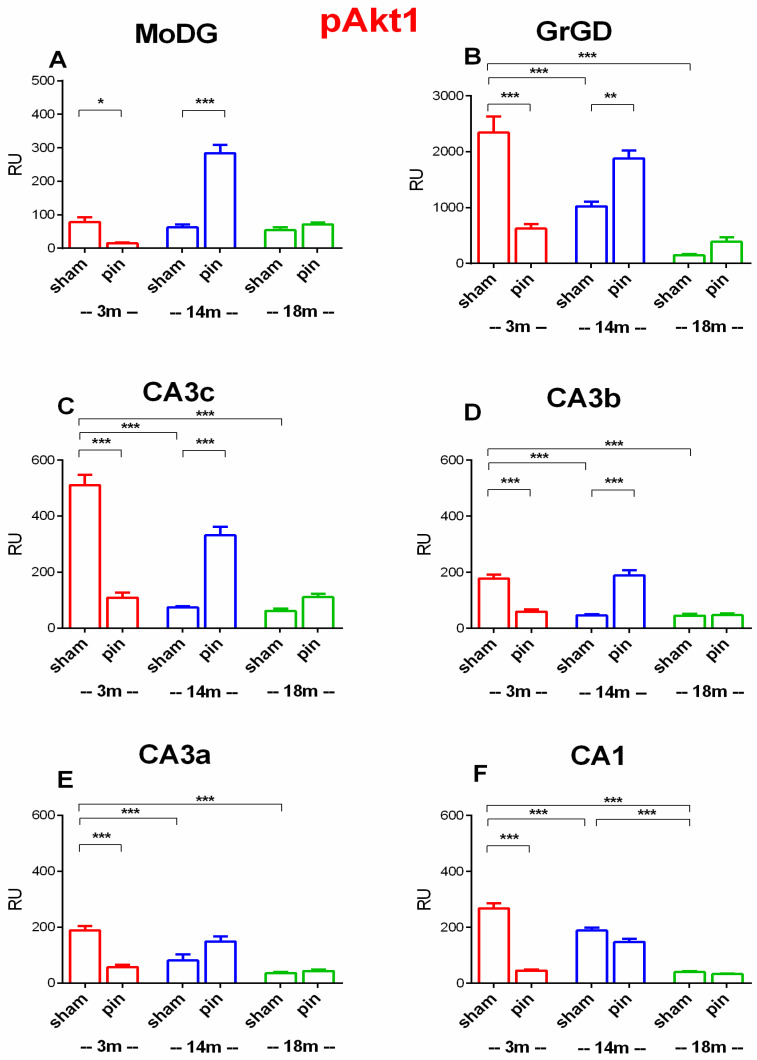
Effect of pinealectomy on pAkt1 protein expression in the dHipp, including the MoDG, GrDG, PoDG, CA3c, CA3b, CA3a, CA2, and CA1 regions. Bars show mean ± SEM for sham and pinealectomized (pin) rats at 3 months (red), 14 months (blue), and 18 months (green); number of animals (n = 5–6) per group. MoDG (**A**): * *p* = 0.022, 3-month-old pin vs. matched sham rats; *** *p* < 0.001, 14- and 18-month-old pin vs. matched sham rats. GrDG (**B**): *** *p* < 0.001, 14- and 18-month-old sham vs. 3-month-old sham rats; *** *p* < 0.001, 3-month-old pin vs. matched sham rats; ** *p* = 0.010, 14-month-old pin vs. matched sham. CA3c (**C**): *** *p* < 0.001, 14- and 18-month-old sham vs. 3-month-old sham rats; *** *p* < 0.001, 3- and 14-month-old pin vs. matched groups. CA3b (**D**): *** *p* < 0.001, 14- and 18-month-old sham vs. 3-month-old sham rats; *** *p* < 0.001, 3- and 14-month-old pin vs. matched groups. CA3a (**E**): *** *p* < 0.001, 14- and 18-month-old sham vs. 3-month-old sham rats; *** *p* < 0.001, 3-month-old pin vs. matched group. CA1 (**F**): *** *p* < 0.001, 14- and 18-month-old sham vs. 3-month-old sham rats; *** *p* < 0.001, 18-month-old sham vs. 14-month-old sham rats; *** *p* < 0.001, 3-month-old pin vs. matched group.

**Figure 4 ijms-26-08093-f004:**
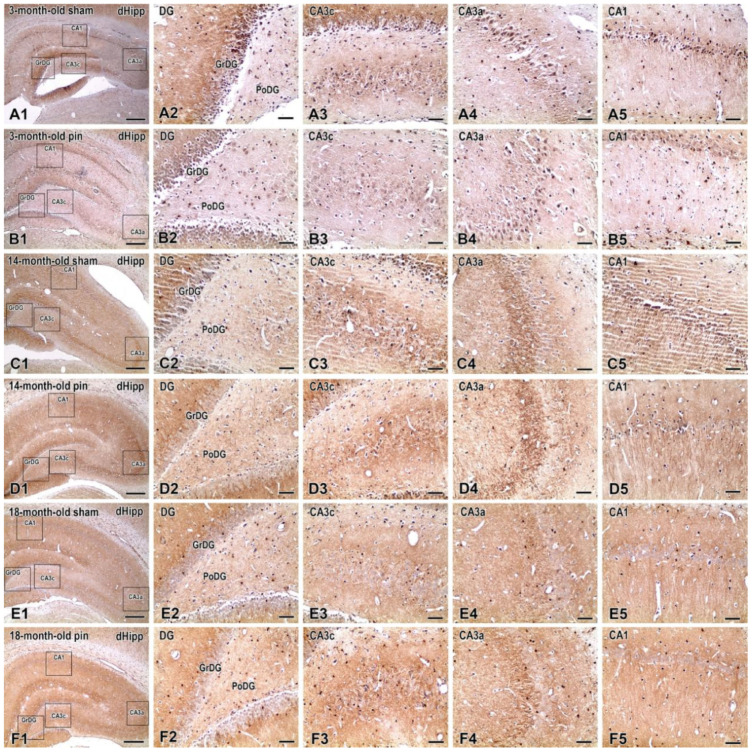
Immunohistochemical detection of NF-κB (p100/p52) in the dHipp of sham-operated and pinealectomized rats at 3, 14, and 18 months of age. Low-magnification overviews (**A1**–**F1**) show hippocampal cytoarchitecture and subregion placement. Higher-magnification images illustrate NF-κB immunoreactivity in the GrDG (**A2**–**F2**), CA3c (**A3**–**F3**), CA3a (**A4**–**F4**), and CA1 (**A5**–**F5**). Immunolabeling was visualized with DAB (brown) and hematoxylin counterstaining. A visible increase in the number of NF-κB-positive cells is evident in the GrDG and CA3c of 14-month-old pinealectomized rats (**D2**,**D3**), with nuclear staining apparent in a subset of cells. Other regions and groups showed fewer labeled cells, with predominantly cytoplasmic immunoreactivity. Scale bars = 500 µm (**A1**,**B1**,**C1**,**D1**,**E1**,**F1**), 100 μm (**A2**–**A5**,**B2**–**B5**,**C2**–**C5**,**D2**–**D5**,**E2**–**E5**,**F2**–**F5**).

**Figure 5 ijms-26-08093-f005:**
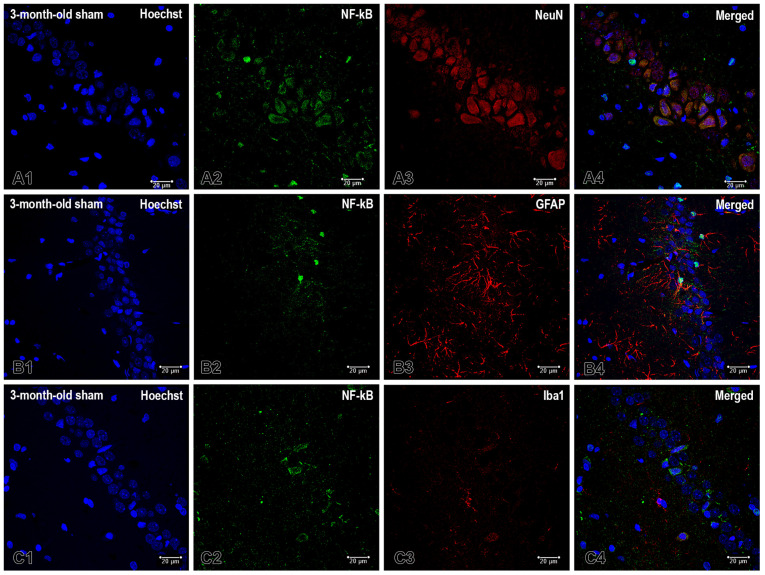
The cell-type-specific localization of NF-κB in the hippocampal formation of 3-month-old sham-operated rats was revealed by triple-label immunofluorescence. Representative micrographs from the pyramidal cell layer (stratum pyramidale) in CA1 show Hoechst (blue), NF-κB (green), and cell-type markers (red). Merged panels illustrate colocalization (**A1**–**A4**). (**A1**–**A4**) show NF-κB with the neuronal marker NeuN. NF-κB labeling is prominent in CA1 pyramidal neurons and frequently overlaps with NeuN, appearing yellow/orange in the merged image. Perinuclear and intranuclear puncta are evident, consistent with NF-κB distribution across the cytoplasm and nucleus. (**B1**–**B4**) NF-κB with the astrocytic marker GFAP: GFAP-positive processes are abundant around the stratum pyramidale and extend into the stratum radiatum. However, they exhibit only sparse spatial overlap with the NF-κB channel. (**C1**–**C4**) show NF-κB with the microglial marker Iba1. Ramified Iba1-positive profiles are present, but they show minimal colocalization with NF-κB within the stratum pyramidale. The images shown are from 3-month-old rats that underwent a sham operation, and the exposure settings were identical across channels. The same qualitative cellular pattern was observed in other experimental groups. Scale bar = 20 μm.

**Figure 6 ijms-26-08093-f006:**
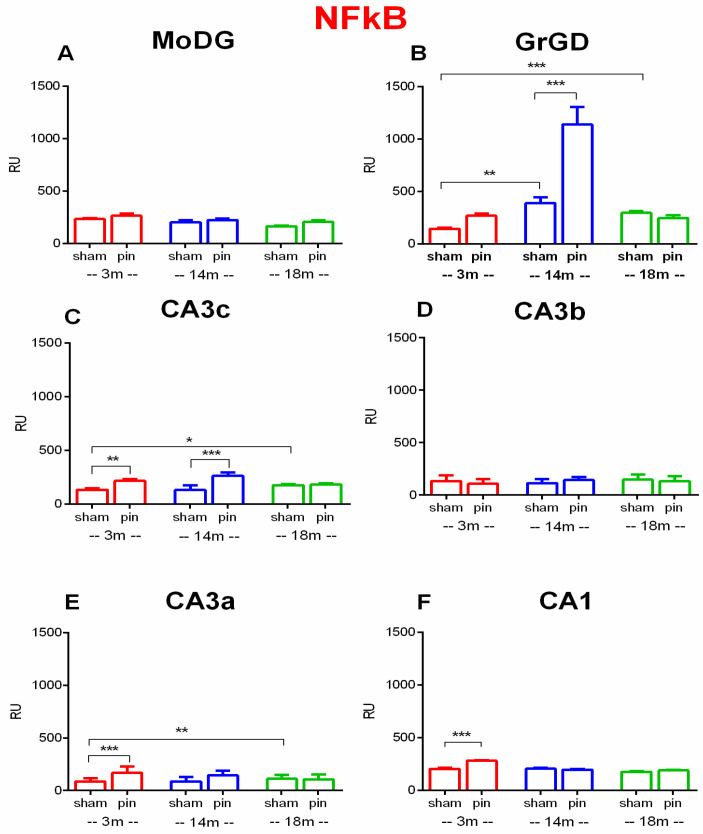
Effect of pinealectomy on NF-kB expression in the dHipp, including the MoDG, GrDG, PoDG, CA3c, CA3b, CA3a, CA2, and CA1 regions. Bars show mean ± SEM for sham and pinealectomized (pin) rats at 3 months (red), 14 months (blue), and 18 months (green); number of animals (n = 5–6) per group. MoDG (**A**): *p* > 0.05, among sham 3,14,18 groups and pin vs. sham groups; GrDG (**B**): ** *p* < 0.01, 14-month-old sham vs. 3-month-old rats; *** *p* < 0.001, 18-month-old sham vs. 3-month-old rats; *** *p* < 0.001, 14-month-old pin rats vs. matched sham controls. CA3c (**C**): * *p* < 0.05, 18-month-old sham vs. 3-month-old sham rats; ** *p* < 0.01, 3-month-old pin rats vs. matched sham; *** *p* < 0.001, 14-month-old pin vs. matched sham rats; CAb (**D**): *p* > 0.05, among sham 3,14,18 groups and pin vs. sham groups; CA3a (**E**): ** *p* < 0.01, 18-month-old sham vs. 3-month-old rats; *** *p* < 0.001, 3-month-old pin rats vs. matched sham. CA1 (**F**): *** *p* < 0.001, 3-month-old pin rats vs. matched sham controls.

**Figure 7 ijms-26-08093-f007:**
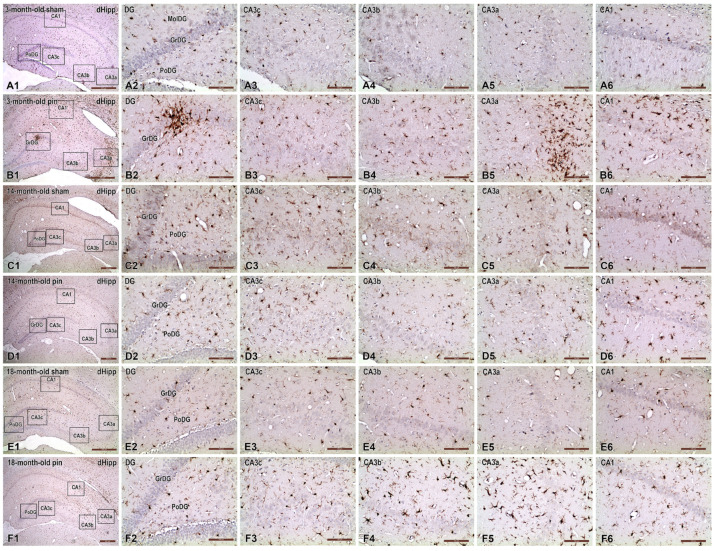
Representative photomicrographs demonstrating age-related and pinealectomy-induced alterations in microglial cells within the dHipp of male rats. Coronal sections from 3-, 14-, and 18-month-old sham-operated (sham) and pinealectomized (pin) rats were immunostained for Iba1, a marker of microglia. Panels (**A1**–**F1**) illustrate low-magnification overviews, identifying specific hippocampal subregions analyzed in detail: MoDG, GrDG, and PoDG layers of the DG, along with hippocampal subfields CA3c, CA3b, CA3a, and CA1. Panels (**A2**–**F6**) represent higher magnification images, clearly depicting microglial morphology and distribution across the respective hippocampal areas under different experimental conditions. Scale bars = 500 µm (**A1**,**B1**,**C1**,**D1**,**E1**,**F1**), 100 μm (**A2**–**A6**,**B2**–**B6**,**C2**–**C6**,**D2**–**D6**,**E2**–**E6**,**F2**–**F6**).

**Figure 8 ijms-26-08093-f008:**
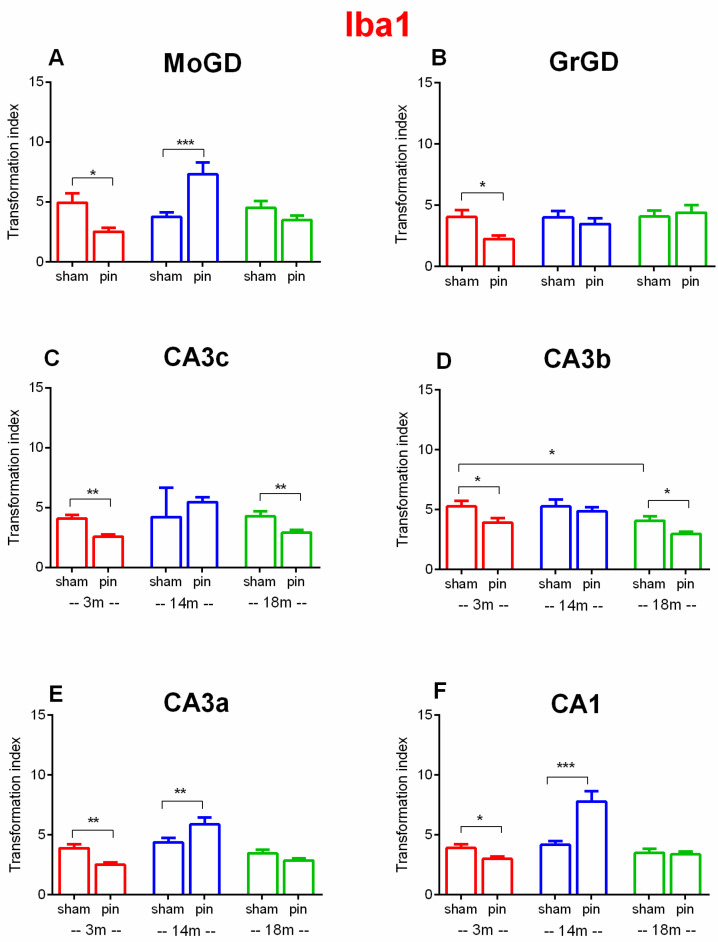
Effect of pinealectomy on Iba1 expression in the dHipp, including the DG, CA3c, CA3b, CA3a and CA1 regions. Transformation index of Iba1^+^ microglia across hippocampal subfields (**A**) MoDG, (**B**) GrDG, (**C**) CA3c, (**D**) CA3b, (**E**) CA3a, and (**F**) CA1. Bars show mean ± SEM for sham and pinealectomized (pin) rats at 3 months (red), 14 months (blue), and 18 months (green); number of animals (n = 5–6) per group. TI was computed on individually segmented Iba1-positive microglia (single-cell morphometry) as TI = P^2^/(4πA), where *p* is perimeter and A is cell area (higher values indicate more ramified morphology). MoDG (**A**): * *p* = 0.021, 3-month-old pin rats vs. matched sham controls. *** *p* < 0.001, 14-month-old pin rats vs. matched sham controls. GrDG: (**B**) * *p* = 0.011, 3-month-old pin rats vs. matched sham controls. CA3c (**C**): ** *p* = 0.003, 3- and 18-month-old pin rats vs. matched sham controls. CA3b (**D**): * *p* = 0.017, 18-month-old sham vs. 3-month-old sham rats; * *p* = 0.032, 3-month-old pin rats vs. matched sham controls; * *p* = 0.016, 18-month-old pin rats vs. matched sham rats. CA3a (**D**): ** *p* = 0.006, 3-month-old pin rats vs. matched sham control; ** *p* = 0.003, 14-momth-old pin rats vs. matched controls. CA1 (**E**) * *p* = 0.0108, 3-month-old pin rats vs. matched sham control; *** *p* < 0.001, 14-momth-old pin rats vs. matched controls.

**Figure 9 ijms-26-08093-f009:**
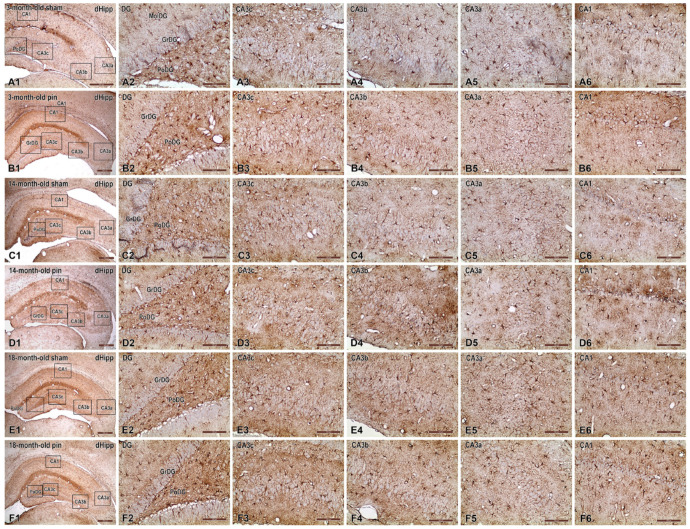
Representative photomicrographs of GFAP immunostaining in the dHipp of sham-operated and pinealectomized (pin) rats at different ages (3, 14, and 18 months). Panels (**A1**–**F1**) illustrate low-magnification images showing regional organization of hippocampal layers and subfields examined (MoDG, GrDG, PoDG, CA3c, CA3b, CA3a, and CA1). High-magnification images (panels (**A2**–**F6**)) display the detailed astrocytic morphology and GFAP expression levels within the respective hippocampal subfields for each experimental condition and age group. Astrocytic activation and hypertrophy progressively increase with age, and are further exacerbated by pinealectomy, particularly notable in the DG and CA3 subfields. Scale bars = 500 µm (**A1**,**B1**,**C1**,**D1**,**E1**,**F1**), 100 μm (**A2**–**A6**,**B2**–**B6**,**C2**–**C6**,**D2**–**D6**,**E2**–**E6**,**F2**–**F6**).

**Figure 10 ijms-26-08093-f010:**
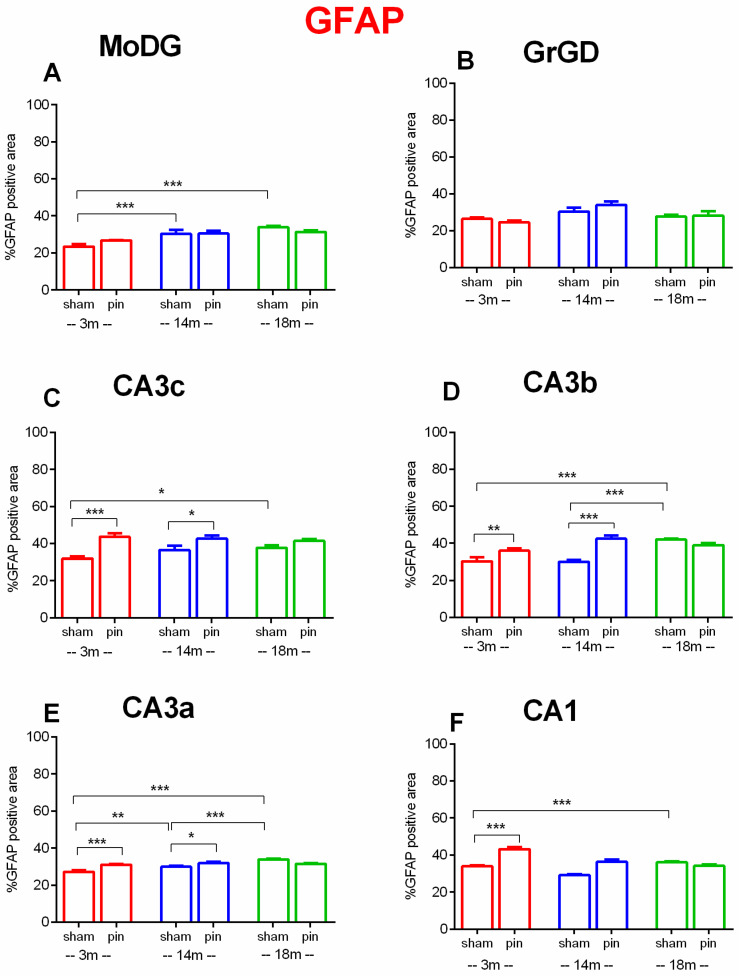
Effect of pinealectomy on GFAP expression in the dHipp, including the MoDG, GrDG, CA3c, CA3b, CA3a, and CA1 regions. Bars show mean ± SEM for sham and pinealectomized (pin) rats at 3 months (red), 14 months (blue), and 18 months (green); number of animals (n = 5–6) per group. MoDG: (**A**): *** *p* < 0.001, 14-month-old sham vs. 3-month-old sham rats; *** *p* < 0.001, 18-month-old sham vs. 3-month-old sham rats; GrDG (**B**): *p* > 0.05, among sham 3,14,18 groups and pin vs. sham groups; *p* > 0.05, among sham 3,14,18 groups and pin vs. sham groups; CA3c: (**C**): * *p* = 0.031, 18-month-old sham vs. 3-month-old sham rats; *** *p* < 0.001, 3-month-old pin vs. matched sham rats; * *p* = 0.03, 14-month-old pin vs. matched sham rats. CA3b: (**D**): *** *p* < 0.001, 18-month-old sham vs. 3-month-old sham rats; *** *p* < 0.001, 18-month-old sham vs. 14-month-old sham rats.** *p* = 0.008, 3-month-old pin vs. matched sham rats; *** *p* < 0.001, 14-month-old pin vs. matched sham rats. CA3a: (**E**) ** *p* = 0.004, 14-month-old sham vs. 3-month-old sham rats; *** *p* < 0.001, 18-month-old sham rats vs. 14-month-old sham rats; *** *p* < 0.001, 18-month-old sham rats vs. 3-month-old sham rats; *** *p* < 0.001, 3-month-old pin vs. matched sham rats; * *p* = 0.039, 14-month-old pin vs. matched sham rats; CA1: (**F**): *** *p* < 0.001, 18-month-old sham vs. 3-month-old sham rats; *** *p* < 0.001, 3-month-old pin vs. matched sham rats.

**Table 1 ijms-26-08093-t001:** A summary of how age affects neuroinflammation due to pinealectomy in the neurons and glia in the dHipp, as indicated by Akt and NFkB markers in neurons and glial cells.

Inflammatory Signaling	% Change Pin 3-Months	% Change Pin 14-Months	% Change Pin 18-Months	Glial Markers	% Change Pin 3-Months	% Change Pin 14-Months
**Akt**				**Iba1**		
MoDG	81%↓	358%↑	─	MoDG	49%↓	95%↑
GrDG	73%↓	84%↑	─	GrDG	45%↓	─
CA3c	79%↓	350%↑	─	CA3c	37%↓	─
CA3b	66%↓	307%↑	─	CA3b	26%↓	─
CA3a	70%↓	─	─	CA3a	35%↓	34%↑
CA1	83%↓	─	─	CA1	23%↓	87%↑
**NFkB**				**GFAP**		
MoDG	─	─	─	MoDG	─	─
GrDG	─	193%↑	─	GrDG	─	─
CA3c	62%↑	99%↑	─	CA3c	37%↑	17%↑
CA3b	─	─	─	CA3b	18%↑	41%↑
CA3a	98%↑	─	─	CA3a	14%↑	6%↑
CA1	38%↑	─	─	CA1	27%↑	─

## Data Availability

The data that support the findings of this study are available from the corresponding author upon reasonable request.

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
