# Peer review of "Pinealectomy-Induced Neuroinflammation Varies with Age in Rats"

_ijms, 2025, doi:10.3390/ijms26168093_

Round 1

Reviewer 1 Report

Comments and Suggestions for Authors

General comments:

In this manuscript the authors describe a comprehensive investigation of the expression of four markers in six specific regions of rat hippocampus comparing the effect of pinealectomy performed in rats of three different ages.  The data is generally clearly presented and has been analysed both qualitatively and quantitatively. The information acquired in this work is certainly a useful contribution to this field. There are a few points that should be clarified to enhance the interpretation of the results

  1. The authors do not seem to report anywhere the time lapse between when the pinealectomy is performed and when the animals are sacrificed for the brain analysis. Presumably this gap is the same for all ages of the animals, but it could be an important point of discussion. For example, could it be that older animal require longer time to show an effect of pinealectomy?
  2. The authors compare the results of pinealectomy to sham operated animals, which is of course standard practice. However, it would be useful to have a statement whether the authors have noticed anything unusual in the pattern of protein expression/phosphorylation in the sham operated animal compared to naïve animals. Also, in reference to the point above, if the animals are sacrificed at a relative short time point after the sham or real surgery they may have be responding with a stress reaction that could be compensated or augmented by the lack of melatonin.
  3. The authors report the statistical value of both the overall effect of pinealectomy and of post-hoc analysis of the treatments at different ages. While the post hoc analysis provides very interesting information, it is more difficult to interpret the significance of the overall statistical change over multiple ages when the author clearly show that there are both increases and decreases of expression / phosphorylation.   It would be useful for the author to clarify how they interpret the fact that for example  there is an overall statistical effect of surgery in the MoDG and not in the GrDG when they show the similar bi-phasic changes
  4. The authors’ work has focused on two markers related to inflammation Akt and NF-kB, and two cellular markers Iba1 and GAFP respectively for microglia and astrocytes. However, as no double staining for any marker is reported it would be interesting for the authors to comment in which cells they believe Akt and NF-kB expression is altered by age and surgery. The authors could consider whether it would be of value to present in a single panel the surgery-induced percentage change of expression of inflammatory markers and cells markers as it may give some clues about which cells are affected
  5. In the discussion it would be useful for the authors to identify what are the some of the limitation of their study (e.g. pinealectomy may cause lack of  other mechanisms rather than just melatonin depletion)  and what are the remaining gap of knowledge in this field and how they could be addressed.

Minor  corrections:

Line 46:   ‘vicious circle’ is not a commonly used scientific expression, ‘self perpetuating’ or ‘positive feed-back loop’ may be better

Lines 47-51 References needed for the three statements

Line 67  the word ‘attached’ could be substituted with ‘targeted’

Line 71  “...melatonin have....should be “..melatonin has...”

Line 135 Formatting : the figure legend should not be separated  from the title

Line 195 n number is reported as 5-6. It should be clarified here and in all other similar figures  or in the methods whether this n number refer to  individual slices from different animals or whether any sequential slice has been used

Line 197  the authors define the meaning of the number of stars with respect of  the p value for each of the figure of the same panel . This seems unnecessarily repetitive as they are all the same. The same applies to all other similar figures

Line 224 it may be useful to indicate with arrows in one picture what is referred to as diffuse cytoplasmic staining vs. nuclear staining as there is no nuclear marker staining.

Line 274 From a visual observation of the figure there does not seem to be any difference between sham and pin. rats for 18 months in CA3c. The authors may want to check the data

Line 622 Something wrong with the sentence, is it mice or rats?

Line 624 Initial part of the sentence needs adjustment

Author Response

Reviewer #1

Point #1: The authors do not seem to report anywhere the time lapse between when the pinealectomy is performed and when the animals are sacrificed for the brain analysis. Presumably this gap is the same for all ages of the animals, but it could be an important point of discussion. For example, could it be that older animal require longer time to show an effect of pinealectomy?

Response: We are thankful for this relevant note from the Reviewer and fully agree that this missing information is necessary for the interpretation of pin- and age-associated changes. In the new version, this information was inserted in the Method section. The euthanization (after perfusion) of all groups was conducted a month after the surgical manipulation. In our previous study, we reported that adaptive changes occur in specific parameters 3 months after pinealectomy (http://dx.doi.org/10.1016/j.bbr.2015.12.043). In several other manuscripts, we’ve reported that 1 month after pinealectomy is a crucial period for changes in various signaling molecules important for plasticity in the hippocampus (https://doi.org/10.1016/j.neuint.2025.105960).

Point #2 The authors compare the results of pinealectomy to sham operated animals, which is of course standard practice. However, it would be useful to have a statement whether the authors have noticed anything unusual in the pattern of protein expression/phosphorylation in the sham operated animal compared to naïve animals. Also, in reference to the point above, if the animals are sacrificed at a relative short time point after the sham or real surgery they may have be responding with a stress reaction that could be compensated or augmented by the lack of melatonin.

Response: We did not include data from naive controls that did not undergo surgery due to project constraints set by the local Ethical Commission, which limited the number of groups we could use. However, when we compared data from sham groups with results from naive rats used in our previous experiments, we found no significant differences in neuroinflammation levels. Additionally, as we mentioned in response to Point #1, all rats were euthanized at least a month after the surgery, ensuring that the stress-related effects from the surgical manipulations were minimal.

Point #3 The authors report the statistical value of both the overall effect of pinealectomy and of post-hoc analysis of the treatments at different ages. While the post hoc analysis provides very interesting information, it is more difficult to interpret the significance of the overall statistical change over multiple ages when the author clearly show that there are both increases and decreases of expression / phosphorylation.   It would be useful for the author to clarify how they interpret the fact that for example  there is an overall statistical effect of surgery in the MoDG and not in the GrDG when they show the similar bi-phasic changes

Response: The analysis of the dentate gyrus (DG) revealed significant differences in Akt expression between the sham and pin groups in both the molecular layer of the DG (MoDG) and the granule layer of the DG (GrDG). Regarding NFkB expression, no age-related effects were observed in the MoDG; however, such effects were evident in the GrDG. This suggests that there are functional differences between the neurons in the two layers of the DG. In contrast, glial cells were primarily affected in the MoDG, but not in the GrDG.

Point# 4 The authors’ work has focused on two markers related to inflammation Akt and NF-kB, and two cellular markers Iba1 and GAFP respectively for microglia and astrocytes. However, as no double staining for any marker is reported it would be interesting for the authors to comment in which cells they believe Akt and NF-kB expression is altered by age and surgery.

Response: We are thankful for this relevant remark. In the new version of the manuscript, we used triple stained immunofluorescence in the CA1 subfield of the dHipp of 3-month-old sham rats. Two additional representative confocal micrographs are inserted in the text (Fig.2, page 5, Fig. 4, page 11). These results are also described and discussed in the text.

Point# 5 The authors could consider whether it would be of value to present in a single panel the surgery-induced percentage change of expression of inflammatory markers and cells markers as it may give some clues about which cells are affected

Response: We are thankful for this valuable advice! Considering the precise distribution of Akt and NF-κB after double fluorescence staining, we have inserted a table in the revised manuscript with calculated changes of pin vs. sham for each age, expressed as a percentage (increase or decrease) in specific cell types (neurons or glia) (Table 1 in the Discussion section).

Point# 6 In the discussion it would be useful for the authors to identify what are the some of the limitation of their study (e.g. pinealectomy may cause lack of  other mechanisms rather than just melatonin depletion)  and what are the remaining gap of knowledge in this field and how they could be addressed.

Response: We are thankful for this suggestion, which would improve the quality of the manuscript. Following this advice of the Reviewer an additional subsection was inserted before “The conclusion” entitled “5. Limitation and future direction”.

Point # 7 Minor  corrections: Line 46:   ‘vicious circle’ is not a commonly used scientific expression, ‘self perpetuating’ or ‘positive feed-back loop’ may be better

Response: Corrected.

Point # 8 Lines 47-51 References needed for the three statements

Response: Following the recommendation of the Reviewer three additional references were inserted supporting the three statements.

Point #9 Line 67  the word ‘attached’ could be substituted with ‘targeted’

Response: Corrected.

Point# 10 Line 71  “...melatonin have....should be “..melatonin has...”

Response: Corrected.

Point # 11 Line 135 Formatting : the figure legend should not be separated  from the title

Response: Corrected.

Point # 12 Line 195 n number is reported as 5-6. It should be clarified here and in all other similar figures  or in the methods whether this n number refer to  individual slices from different animals or whether any sequential slice has been used

Response: n = 5-6 refers to the number of animals. For clarity, we edited the text when it was mentioned.

Point # 13 Line 197  the authors define the meaning of the number of stars with respect of  the p value for each of the figure of the same panel . This seems unnecessarily repetitive as they are all the same. The same applies to all other similar figures

Response: We understand that the Reviewer advises simplifying the text by converting detailed descriptions for each figure into a single panel. However, each figure in a panel has different descriptions depending on the post hoc analyses. If we skip this detailed description in the text and instead include it in the figures, it could distort the precise analysis. Therefore, we preferred to simplify the reading for easier understanding of the results

Point # 14 Line 224 it may be useful to indicate with arrows in one picture what is referred to as diffuse cytoplasmic staining vs. nuclear staining as there is no nuclear marker staining. 

Response: We appreciate the reviewer’s suggestion. To avoid visual clutter and because chromogenic bright-field images lack a nuclear marker, we did not annotate the bright-field panel with arrows. Subcellular assignment (nuclear vs. cytoplasmic) is based exclusively on our immunofluorescence data (Figure 4), where nuclei are counterstained with Hoechst and merged channels reveal intranuclear puncta (Figure 4B4) and perinuclear/extranuclear signal (Figure 4A4).

This photograph is from Figure 4A1-A4.

This image clearly shows the nuclear localization of NF-kB, as there is colocalization with the blue nuclear marker Hoechst.

Point # 15 Line 274 From a visual observation of the figure there does not seem to be any difference between sham and pin. rats for 18 months in CA3c. The authors may want to check the data

Response: We agree this note. This was a technical error that was corrected.

Point # 15 Line 622 Something wrong with the sentence, is it mice or rats?

Response: Corrected. We cited the work of Pierpaoli and Bulian’s (2005) conducted on mice there and compared their data to our results in rats in the next sentence.

Point # 16 Line 624 Initial part of the sentence needs adjustment

Response: The two sentences were united with an idea to sounds clearly.

Reviewer 2 Report

Comments and Suggestions for Authors

Review of the Manuscript: "Pinealectomy-induced neuroinflammation varies with age in rats” by Atanasova et al.

The manuscript by Atanasova et al. investigates how pinealectomy influences neuroinflammatory protein expression across different age groups in rats. Melatonin, a fundamental homeostatic hormone, is known to regulate various age-related physiological processes. This study highlights melatonin’s potential regulatory role in hippocampal neuroinflammation, which may significantly impact age-related changes in hippocampal function.

The authors utilize an appropriate animal model and apply conventional immunohistochemistry techniques to present their findings. However, the data are presented in a largely qualitative or semi-quantitative manner (i.e., cell counting within selected ROIs) rather than through objective, automated quantitative methods that could enhance the robustness of the results. Despite these limitations, the study addresses an important topic and provides meaningful insights. I recommend acceptance with major revisions as outlined below.

Major Points

  1. Clarity on pAkt Expression (Section 2.1):
    It is unclear which cell types express pAkt in the presented results. The authors should include at least one representative double immunofluorescence image to identify the cellular sources of pAkt expression. Additionally, if the antibody used is specific to the phosphorylated form of Akt (pAkt), the terminology throughout the manuscript must be consistent—use "pAkt" instead of "Akt" wherever appropriate.
  2. Definition of RU in Figures 2, 4, 6, and 8:
    The method for determining Relative Units (RU) in the figures is not clearly described. Were the values derived from cell-based intensity measurements using ImageJ (e.g., average intensity per cell), or from global intensity values across the entire ROI? The authors should clearly explain the quantification method in the Materials and Methods section.
  3. Discrepancy in RU Values (Figure 2b):
    In Figure 2b, RU values range between 2000–3000, whereas in the other figures, values are closer to ~200. This discrepancy is unexplained and potentially misleading. The authors need to clarify whether this is due to differences in staining protocols, normalization methods, or units used for different markers.
  4. Regional Variability in Hippocampal Expression:
    The manuscript would benefit from a more thorough discussion of why different hippocampal regions exhibit distinct expression patterns for pAkt, NFκB, Iba1, and GFAP. Consider possible anatomical, cellular, or functional differences within hippocampal subfields that might explain this variability.
  5. NFκB Cellular Localization (Section 2.2):
    Similar to the pAkt data, it is not specified which cell types express NFκB. Since NFκB can be expressed by microglia, astrocytes, or infiltrating immune cells, the authors should use double immunofluorescence labeling to identify the expressing cell population.
  6. Quantification of Microglial Morphological Changes (Section 2.3):
    The manuscript states that microglial shape changes with age and pinealectomy, but this is not quantified. The authors should calculate a Transformation Index (Ti) as described in Fujita et al. (1996, Glia 18(4): 269–281) or Szabó et al. (Neuroscience 241: 280–295) using ImageJ or similar software to provide quantitative support.
  7. Astrocytic Morphology Quantification:
    Similar to microglia, the reported morphological changes in GFAP-positive astrocytes should be quantified. Suggested metrics include branch point number, average branch length, or Sholl analysis, using ImageJ or other appropriate image analysis tools.

Conclusion

This study addresses a significant topic and is based on a sound experimental model. However, several important issues require clarification and additional analysis, particularly regarding the cellular localization of immunostained markers and the quantification of morphological changes. Addressing the points above will substantially strengthen the manuscript. I therefore recommend acceptance after major revisions.

Review of the Manuscript: "Pinealectomy-induced neuroinflammation varies with age in rats” by Atanasova et al.

Author Response

Point#1: Clarity on pAkt Expression (Section 2.1):
It is unclear which cell types express pAkt in the presented results. The authors should include at least one representative double immunofluorescence image to identify the cellular sources of pAkt expression. Additionally, if the antibody used is specific to the phosphorylated form of Akt (pAkt), the terminology throughout the manuscript must be consistent—use "pAkt" instead of "Akt" wherever appropriate.

Response: Following the Reviewer note, we edited the text where Akt was mentioned by replacing Akt with pAkt.

Point#2: Definition of RU in Figures 2, 4, 6, and 8:
The method for determining Relative Units (RU) in the figures is not clearly described. Were the values derived from cell-based intensity measurements using ImageJ (e.g., average intensity per cell), or from global intensity values across the entire ROI? The authors should clearly explain the quantification method in the Materials and Methods section.

Response: We apologize for the ambiguity. In all figures, Relative Units (RU) denote cell density, defined as the number of immunopositive cells per 1 mm² within the predefined ROI. RU are not derived from intensity measurements (neither per-cell nor global). Positivity is determined by segmentation of the DAB channel with a fixed, batch-specific threshold and morphological inclusion criteria; counts are then divided by the calibrated ROI area. For GFAP we report two complementary readouts: RU (cells·mm⁻²) when quantifying GFAP-positive astrocytic somata, and % GFAP-positive area when assessing astroglial coverage.

Point#3: Discrepancy in RU Values (Figure 2b):
In Figure 2b, RU values range between 2000–3000, whereas in the other figures, values are closer to ~200. This discrepancy is unexplained and potentially misleading. The authors need to clarify whether this is due to differences in staining protocols, normalization methods, or units used for different markers.

Response: The discrepancy noted is not due to differences in staining protocols or calculation methods, as these are consistent across all studied structures. Instead, it arises from variations in cell density across different layers of the dentate gyrus (DG), with the granular layer of the dentate gyrus (GrDG) having a higher cell density compared to the molecular layer of the dentate gyrus (MoDG).

Point#4: Regional Variability in Hippocampal Expression:
The manuscript would benefit from a more thorough discussion of why different hippocampal regions exhibit distinct expression patterns for pAkt, NFκB, Iba1, and GFAP. Consider possible anatomical, cellular, or functional differences within hippocampal subfields that might explain this variability.

Response: We are thankful for this relevant suggestion. We included additional paragraph in the Conclusion.

Point#5: NFκB Cellular Localization (Section 2.2):
Similar to the pAkt data, it is not specified which cell types express NFκB. Since NFκB can be expressed by microglia, astrocytes, or infiltrating immune cells, the authors should use double immunofluorescence labeling to identify the expressing cell population.

Response: We are thankful for this relevant remark. In the new version of the manuscript, we used triple stained immunofluorescence in the CA1 subfield of the dHipp of 3-month-old sham rats. Two additional representative confocal micrographs are inserted in the text (Fig.2, page 5, Fig. 4, page 11). These results are also described and discussed in the text.

Point#6: Quantification of Microglial Morphological Changes (Section 2.3):
The manuscript states that microglial shape changes with age and pinealectomy, but this is not quantified. The authors should calculate a Transformation Index (Ti) as described in Fujita et al. (1996, Glia 18(4): 269–281) or Szabó et al. (Neuroscience 241: 280–295) using ImageJ or similar software to provide quantitative support.

Response: We are thankful for this relevant suggestion. We used the recommended method to measure and calculate TI for microglia. In the new version of the manuscript all data for IBA1 are given by TI.

Point#7: Astrocytic Morphology Quantification:
Similar to microglia, the reported morphological changes in GFAP-positive astrocytes should be quantified. Suggested metrics include branch point number, average branch length, or Sholl analysis, using ImageJ or other appropriate image analysis tools.

Response:  We appreciate the suggestion to quantify astrocyte morphology (branch points, average branch length, Sholl analysis). However, our material consists of thin 2D sections acquired as single optical planes. Under these conditions, Sholl- and skeleton-based metrics are prone to systematic bias due to process truncation and sectioning angle, and are recommended primarily for 3D image stacks of individually reconstructed astrocytes. Moreover, GFAP immunolabeling highlights intermediate filaments in main stem processes and does not reliably capture the fine perisynaptic astrocytic processes, leading to an underestimation of branching complexity when applying Sholl/skeletonisation on GFAP images.

Accordingly, our quantification was performed across seven anatomically defined hippocampal ROIs—CA1, CA3a, CA3b, CA3c, the granule layer of the dentate gyrus (GrDG), the molecular layer (MoDG), and the polymorphic layer/hilus (PoDG). Within each ROI, we used the percent GFAP-positive area (area fraction) as our primary quantitative readout of astrocyte reactivity/hypertrophy. This metric is robust for thin sections, insensitive to z-truncation, and is widely used to quantify astrogliosis. We have clarified this rationale in the Methods and report ROI-specific results.

Conclusion

This study addresses a significant topic and is based on a sound experimental model. However, several important issues require clarification and additional analysis, particularly regarding the cellular localization of immunostained markers and the quantification of morphological changes. Addressing the points above will substantially strengthen the manuscript. I therefore recommend acceptance after major revisions.

Round 2

Reviewer 2 Report

Comments and Suggestions for Authors

I accept the answers, and I suggest the article for submission in the reviewed form.